# Deep learning applied to glacier evolution modelling

Jordi Bolibar[1,2], Antoine Rabatel[1], Isabelle Gouttevin[3], Clovis Galiez[4], Thomas Condom[1], and Eric Sauquet[2]

[1]Univ. Grenoble Alpes, CNRS, IRD, G-INP, Institut des Géosciences de l'Environnement (IGE, UMR 5001), Grenoble, France
[2]Irstea, UR RiverLy, Lyon-Villeurbanne, France
[3]Univ. Grenoble Alpes, Université de Toulouse, Météo-France, CNRS, CNRM, Centre d'Études de la Neige, Grenoble, France
[4]Univ. Grenoble Alpes, CNRS, Grenoble INP, LJK, Grenoble, France

**Correspondence:** Jordi Bolibar (jordi.bolibar@univ-grenoble-alpes.fr)

**Abstract.** We present a novel approach to simulate and reconstruct annual glacier-wide surface mass balance (SMB) series based on a deep artificial neural network (*i.e.* deep learning). This method has been included as the SMB component of an open-source regional glacier evolution model. While most glacier models tend to incorporate more and more physical processes, here we take an alternative approach by creating a parameterized model based on data science. Annual glacier-wide SMBs can
be simulated from topo-climatic predictors using either deep learning or Lasso (regularized multilinear regression), whereas the glacier geometry is updated using a glacier-specific parameterization. We compare and cross-validate our nonlinear deep learning SMB model against other standard linear statistical methods on a dataset of 32 French alpine glaciers. Deep learning is found to outperform linear methods, with improved explained variance (up to +64% in space and +108% in time) and accuracy (up to +47% in space and +58% in time), resulting in an estimated $r^2$ of 0.77 and RMSE of 0.51 m.w.e. Substantial nonlinear
structures are captured by deep learning, with around 35% of nonlinear behaviour in the temporal dimension. For the glacier geometry evolution, the main uncertainties come from the ice thickness data used to initialize the model. These results should encourage the use of deep learning in glacier modelling as a powerful nonlinear tool, capable of capturing the nonlinearities of the climate and glacier systems, that can serve to reconstruct or simulate SMB time series for individual glaciers in a whole region for past and future climates.

## 1  Introduction

Glaciers are arguably one of the most important icons of climate change, being climate proxies which can depict the evolution of climate for the global audience (IPCC, 2018). In the coming decades, mountain glaciers will be some of the most important contributors to sea level rise and will most likely drive important changes in the hydrological regime of glaciarized catchments (Beniston et al., 2018; Vuille et al., 2018; Hock et al., 2019). The reduction in ice volume may produce an array of hydrological,
ecological and economic consequences in mountain regions which requires to be properly predicted. These consequences will strongly depend on the future climatic scenarios, which will determine the timing and magnitude for the transition of

hydrological regimes (Huss and Hock, 2018). Understanding these future transitions is key for societies to adapt to future hydrological and climate configurations.

Glacier and hydro-glaciological models can help answer these questions, giving several possible outcomes depending on multiple climate scenarios. (a) Surface mass balance (SMB) and (b) glacier dynamics both need to be modelled to understand glacier evolution on regional and sub-regional scales. Models of varying complexity exist for both processes. In order to model these processes at large scale (*i.e.* on several glaciers at a catchment scale), some compromises need to be made, which can be approached in different ways:

(a) Regarding SMB:

1. Empirical models, like the temperature-index model (e.g. Hock, 2003), simulate glacier SMB through empirical relationships between air temperature and melt and snow accumulation.

2. Statistical or machine learning models describe and predict glacier SMB based on statistical relationships found in data from a selection of topographical and climate predictors (e.g. Martin, 1974; Steiner et al., 2005).

3. Physical and Surface Energy Balance (SEB) models take into account all energy exchanges between the glacier and the atmosphere, and can simulate the spatial and temporal variability of snowmelt and the changes in albedo (e.g. Gerbaux et al., 2005).

(b) Regarding glacier dynamics:

1. Parameterized models do not explicitly resolve any physical processes, but implicitly take them into account using parameterizations, based on statistical or empirical relationships, in order to modify the glacier geometry. This type of models range from very simple statistical models (e.g. Carlson et al., 2014) to more complex ones based on different approaches, such as a calibrated equilibrium-line altitude (ELA) model (e.g. Zemp et al., 2006), a glacier retreat parameterization specific for glacier size groups (Huss and Hock, 2015) or volume/length-area scaling (e.g. Marzeion et al., 2012; Radić et al., 2014).

2. Process-based models, like GloGEMflow (e.g. Zekollari et al., 2019) and OGGM (e.g. Maussion et al., 2019), approximate a number of glacier physical processes involved in ice flow dynamics using the shallow ice approximation.

3. Physics-based models, like the finite elements Elmer/Ice model (e.g. Gagliardini et al., 2013), approach glacier dynamics by explicitly simulating physical processes and solving the full Stokes equations (e.g. Jouvet et al., 2009; Réveillet et al., 2015).

At the same time, the use of these different approaches strongly depend on available data, whose spatial and temporal resolutions have an important impact on the results' quality and uncertainties (e.g., Réveillet et al., 2018). Parameterized glacier dynamics models and empirical and statistical SMB models require a reference or training dataset to calibrate the relationships, which can then be used for projections with the hypothesis that relationships remain stationary in time. On the

contrary, process-based and specially physics-based glacier dynamics and SMB models have the advantage of representing physical processes, but they require larger datasets at higher spatial and temporal resolutions with a consequently higher computational cost (Réveillet et al., 2018). For SMB modelling, meteorological reanalyses provide an attractive alternative to sparse point observations, although their spatial resolution and suitability to complex high-mountain topography are often not good enough for high-resolution physics-based glacio-hydrological applications. However, parameterized models are much more flexible, equally dealing with fewer and coarser meteorological data as well as the state of the art reanalyses, which allows to work at resolutions much closer to glaciers' scale and to reduce uncertainties. The current resolution of climate projections is still too low to adequately drive most glacier physical processes, but the ever-growing datasets of historical data are paving the way for the training of parameterized machine learning models.

In glaciology, statistical models have been applied for more than half a century, starting with simple multiple linear regressions on few meteorological variables (Hoinkes, 1968; Martin, 1974). Statistical modelling has made enormous progress in the last decades, specially thanks to the advent of machine learning. Compared to other fields in geosciences, such as oceanography (e.g., Ducournau and Fablet, 2016; Lguensat et al., 2018), climatology (e.g., Rasp et al., 2018; Jiang et al., 2018) and hydrology (e.g., Marçais and de Dreuzy, 2017; Shen, 2018), we believe that the glaciological community has not yet exploited the full capabilities of these approaches. Despite this fact, a number of studies have taken steps towards statistical approaches. Steiner et al. (2005) pioneered the very first study to use artificial neural networks (ANNs) in glaciology to simulate mass balances of the Grosse Aletschgletscher in Switzerland. They showed that a nonlinear model is capable of better simulating glacier mass balances compared to a conventional stepwise multiple linear regression. Furthermore, they found a significant nonlinear part within the climate/glacier mass balance relationship. This work was continued in Steiner et al. (2008) and Nussbaumer et al. (2012) for the simulation of glacier length instead of mass balances. Later on, Maussion et al. (2015) developed an empirical statistical downscaling tool based on machine learning in order to retrieve glacier surface energy and mass balance (SEB/SMB) fluxes from large-scale atmospheric data. They used different machine learning algorithms, but all of them were linear, which are not necessarily the most suitable for modelling the nonlinear climate system (Houghton et al., 2001). Nonetheless, more recent developments in the field of machine learning and optimization enabled the use of deeper network structures than the 3-layer ANN of Steiner et al. (2005). These deeper ANNs, which remain unexploited in glaciology, allow to capture more nonlinear structures in the data even for relatively small datasets (Ingrassia and Morlini, 2005; Olson et al., 2018).

Here, we present a parameterized regional open-source glacier model: the ALpine Parameterized Glacier Model (ALPGM, Bolibar, 2019). When most glacier evolution models tend to incorporate more and more physical processes in SMB or ice dynamics (e.g., Maussion et al., 2019; Zekollari et al., 2019), ALPGM takes an alternative approach based on data science for SMB modelling and parameterizations for glacier dynamics simulation. ALPGM simulates annual glacier-wide SMB and the evolution of glacier volume and surface area over time scales from a few years to a century at a regional scale. Glacier-wide SMBs are computed using a deep ANN, fed by several topographical and climatic variables, an approach which is compared to different linear methods in the present paper. In order to distribute these annual glacier-wide SMBs and to update the glacier geometry, a refined version of the $\Delta h$ methodology (e.g., Huss et al., 2008) is used, for which we dynamically compute glacier-specific $\Delta h$ functions. In order to validate this approach, we use a case study with 32 French alpine glaciers for which glacier-

wide annual SMBs are available over the period 1984-2014 and 1959-2015 for certain glaciers. High resolution meteorological reanalyses for the same time period are used (SAFRAN, Durand et al., 2009) while the initial ice thickness distribution of glaciers are taken from Farinotti et al. (2019), for which we performed a sensitivity analysis based on field observations.

In the next section, we present an overview of the proposed glacier evolution model framework with a detailed description of the two components used to simulate the annual glacier-wide SMB and the glacier geometry update. Then, a case study using French alpine glaciers is presented, which enables to illustrate an example of application of the proposed framework including a rich dataset, the parameterized functions, as well as the results and their performance. In the end, several aspects regarding machine and deep learning modelling in glaciology are discussed, from which we make some recommendations and draw the final conclusions.

## 2 Model overview and methods

In this section we present an overview of the ALPGM glacier model. Moreover, the two components of this model are presented in detail: the Glacier-wide SMB Simulation component and the Glacier Geometry Update component.

### 2.1 Model overview and workflow

ALPGM is an open-source glacier model coded in Python. The source code of the model is accessible in the project repository (see Code availability). It is structured in multiple files which execute specific separate tasks. The model can be divided into two main components: (1) the Glacier-wide SMB Simulation and (2) the Glacier Geometry Update. The Glacier-wide SMB Simulation component is based on machine learning, taking both meteorological and topographical variables as inputs. The Glacier Geometry Update component generates the glacier-specific parameterized functions and modifies annually the geometry of the glacier (*e.g.* ice thickness distribution, glacier outline) based on the glacier-wide SMB models generated by the Glacier-wide SMB simulation component.

Fig. 1 presents ALPGM's basic workflow. The workflow execution can be configured via the model interface, allowing to run or skip any of the following steps:

1. The meteorological forcings are preprocessed in order to extract the necessary data closest to each glacier's centroid. The meteorological features are stored in intermediate files in order to reduce computation times for future runs, automatically skipping this preprocessing step when the files have already been generated.

2. The SMB machine learning component retrieves the preprocessed climate predictors from the stored files, retrieves the topographical predictors from the multitemporal glacier inventories, and then it assembles the training dataset by combining all the necessary topo-climatic predictors. A machine learning algorithm is chosen for the SMB model, which can be loaded from a previous run or it can be trained again with a new dataset. Then, the SMB model(s) are trained with the full topo-climatic dataset. These model(s) are stored in intermediate files, allowing to skip this step for future runs.

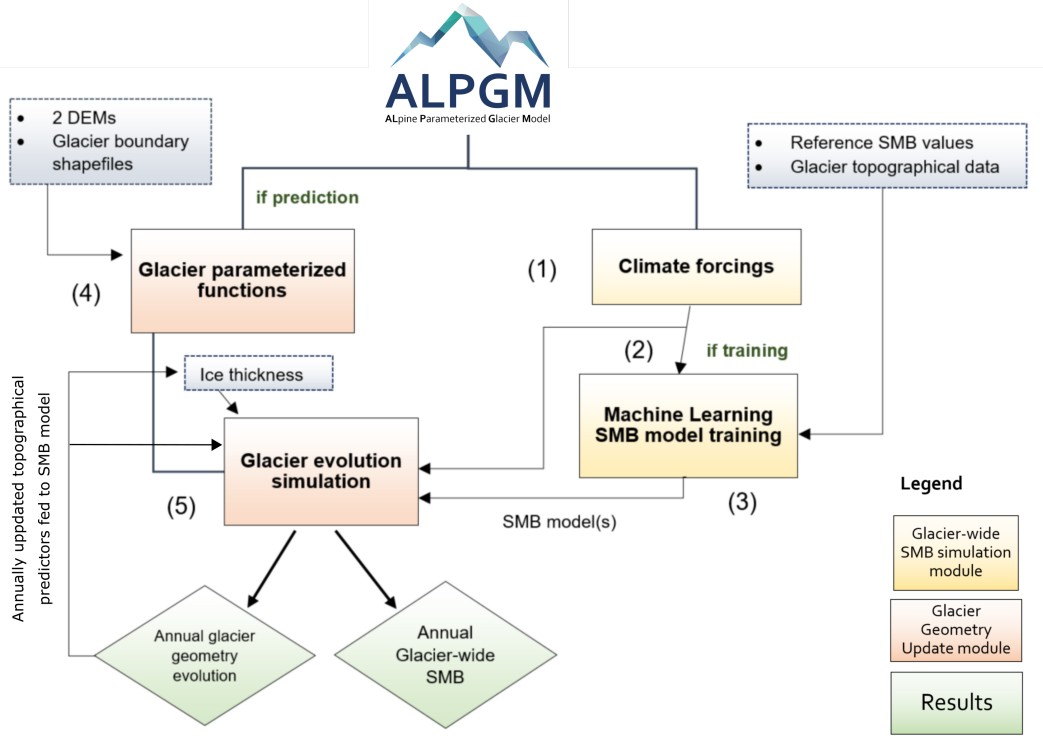

**Figure 1.** ALPGM structure and workflow

3. Performances of the SMB models can be evaluated with a leave-one-glacier-out (LOGO) or a leave-one-year-out (LOYO) cross-validation. This step can be skipped when using already established models. Basic statistical performance metrics are given for each glacier and model, as well as plots with the simulated cumulative glacier-wide SMBs compared to their reference values with uncertainties for each of the glaciers from the training dataset.

4. The Glacier Geometry Update component starts with the generation of the glacier specific parameterized functions, using a raster containing the difference of the two pre-selected digital elevation models (DEMs) covering the study area for two separate dates, as well as the glacier contours. These parameterized functions are then stored in individual files to be used in the final simulations.

5. Once all previous steps have been run, the glacier evolution simulations are launched. For each glacier, the initial ice thickness and DEM rasters and the glacier geometry update function are retrieved. Then, in a loop, for every glacier and year, the topographical data is computed from these raster files. The climate predictors at the glacier's current centroid are retrieved from the climate data (e.g. reanalysis or projections) and with all this data the input topo-climatic data for the glacier-wide SMB model is assembled. Afterwards, the glacier-wide SMB for this glacier and year is simulated, which combined with the glacier-specific geometry update function allows to update the glacier's ice thickness and

DEM rasters. This process is repeated in a loop, therefore updating the glacier's geometry with an annual timestep and taking into account the glacier's morphological and topographical changes in the glacier-wide SMB simulations. For the simulation of the following year's SMB, the previously updated ice thickness and DEM rasters is used to re-compute the topographical parameters, which in turn are used as input topographical predictors for the glacier-wide SMB machine learning model. If all the ice thickness raster pixels of a glacier become zero, the glacier is considered as disappeared and is removed from the simulation pipeline. For each year, multiple results are stored in data files as well as the raster DEM and ice thickness values for each glacier.

## 2.2 Glacier-wide surface mass balance simulation

Annual glacier-wide SMBs are simulated using machine learning. Due to the regional characteristics and specificities of topographical and climate data, this glacier-wide SMB modelling method is, for now, a regional approach.

### 2.2.1 Selection of explanatory topographical and climatic variables

In order to narrow down which topographical and climatic variables best explain glacier-wide SMB in a given study area, a literature review as well as a statistical sensitivity analysis are performed. Typically used topographical predictors are longitude, latitude, glacier slope and mean altitude. As for meteorological predictors, cumulative positive degree days (CPDD), but also mean monthly temperature, snowfall and possibly other variables that influence the surface energy budget are often used in the literature. Examples of both topographic and meteorological predictors can be found in the case study in Sect. 3. A way to prevent biases when making predictions with different climate data is to work with anomalies, calculated as differences of the variable with respect to its average value over a chosen reference period.

For the machine learning training, the relevant predictors must be selected, so we perform a sensitivity study of the annual glacier-wide SMB to topographical and climatic variables over the study training period. This can be performed with individual linear regressions between each variable and glacier-wide SMB data. After identification of the topographical and climatic variables that can potentially explain annual glacier-wide SMB variability for the region of interest, a training dataset is built. An effective way of expanding the training dataset in order to dig deeper into the available data is to combine the climatic and topographical input variables (Weisberg, 2014). Such combinations can be expressed following Eq. (1):

$$SMB_{g,y} = f(\hat{\Omega}, \hat{C}) + \varepsilon_{g,y} \tag{1}$$

Where $\hat{\Omega}$ is a vector of the selected topographical predictors, $\hat{C}$ is a vector with the selected climatic features and $\varepsilon_{g,y}$ is the residual error for each annual glacier-wide SMB value, $SMB_{g,y}$.

Once the training dataset is created, different algorithms $f$ (two linear and one nonlinear, for the case of this study) can be chosen to create the SMB model: (1) OLS (Ordinary Least Squares) all-possible multiple linear regressions; (2) Lasso (Least absolute shrinkage and selection operator) (Tibshirani, 1996); and (3) a deep Artificial Neural Network (ANN). ALPGM uses some of the most popular machine learning Python libraries: StatsModels (Seabold and Perktold, 2010), Scikit-learn (Pedregosa

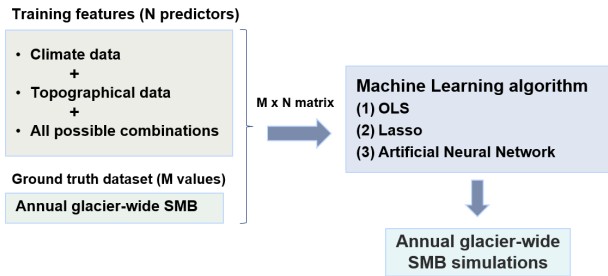

**Figure 2.** Glacier-wide SMB simulation component workflow. Machine learning models are dynamically created based on training data

et al., 2012) and Keras (Chollet, 2015) with a TensorFlow backend. The overall workflow of the machine learning glacier-wide SMB model production in ALPGM is summarized in Fig. 2.

### 2.2.2   All-possible multiple linear regressions

With the ordinary least squares (OLS) all-possible multiple linear regressions, we attempt to find the best subset of predictors in Eq. 1 based on the resulting $r^2$ adjusted, while at the same time avoiding overfitting (Hawkins, 2004) and collinearity, and limiting the complexity of the model. As its name indicates, the goal is to minimize the residual sum of squares for each subset of predictors (Hastie et al., 2009). $n$ models are produced by selecting all possible subsets of $k$ predictors. It is advisable to narrow down the number of predictors for each subset in the search to reduce the computational cost. Models with low performance are filtered out, keeping only models with highest $r^2$ adjusted possible, a variance inflation factor ($VIF$) < 1.2 and a p-value < 0.01/$n$ (in order to ensure the Bonferroni correction). Retained models are combined by averaging their predictions, thereby avoiding the pitfalls related to stepwise single model selection (Whittingham et al., 2006). These criteria ensure that the models explain as much variability as possible, avoid collinearity and are statistically significant.

### 2.2.3   Lasso

The Lasso (Least absolute shrinkage and selection operator) (Tibshirani, 1996) is a shrinkage method which attempts to overcome the shortcomings of the simpler step-wise and all-possible regressions. In these two classical approaches, predictors are discarded in a discrete way, giving subsets of variables which have the lowest prediction error. However, due to its discrete selection, these different subsets can exhibit high variance, which does not reduce the prediction error of the full model. The Lasso performs a more continuous regularization by shrinking some coefficients and setting others to zero, thus producing more interpretable models (Hastie et al., 2009). Because of its properties, it strikes a balance between subset selection (like all-possible regressions) and Ridge regression (Hoerl and Kennard, 1970). All input data is normalized by removing the mean and scaling to unit variance. In order to determine the degree of regularisation applied to the coefficients used in the linear OLS regression, an alpha parameter needs to be chosen using cross-validation. ALPGM performs different types of cross-validations to choose from: the Akaike Information Criterion (AIC), the Bayes Information Criterion (BIC) and a classical cross-validation

with iterative fitting along a regularization path (used in the case study). Alternatively, a Lasso model with Least Angle Regression, also known as Lasso Lars (Tibshirani et al., 2004), can also be chosen with a classical cross-validation.

### 2.2.4 Deep artificial neural network

Artificial neural networks (ANNs) are nonlinear statistical models inspired by biological neural networks (Fausett, 1994; Hastie et al., 2009). A neural network is characterized by: (1) the architecture or pattern of connections between units and the number of layers (input, output and hidden layers); (2) the optimizer: which is the method for determining the weights of the connections between units; and (3) its (usually nonlinear) activation functions (Fausett, 1994). When ANNs have more than one hidden layer (*e.g.* Fig. 3), they are referred to as deep ANNs or deep learning. The description of neural networks is beyond the scope of this study, so for more details and a full explanation please refer to Fausett (1994), Hastie et al. (2009), as well as Steiner et al. (2005, 2008) where the reader can find a thorough introduction to the use of ANNs in glaciology. ANNs gained recent interest thanks to improvements of optimization algorithms allowing the training deep neural networks, that lead to better representation of complex data patterns. As their learnt parameters are difficult to interpret, ANN are adequate tools when the quality of predictions prevails over the interpretability of the model (the latter likely involving causal inference, sensitivity testing or modelling of ancillary variables). This is precisely the case in our study context here, where abundant knowledge about glacier physics further helps choosing adequate variables as input to deep learning. Their ability to model complex functions of the input parameters makes them particularly suitable for modelling complex nonlinear systems such as the climate system (Houghton et al., 2001) and glacier systems (Steiner et al., 2005).

ALPGM uses a feedforward fully-connected ANN (Fig. 3). In such an architecture, the processing units - or neurons - are grouped into layers where all the units of a given layer are fully connected to all units of the next layer. The flow of information is directional, from the input layer (*i.e.*. in which each neuron corresponds to one of the N explanatory variables) to the output neuron (*i.e.*. corresponding to the target variable of the model, the SMB). For each connection of the ANN, weights are initialized in a random fashion following a specific distribution (generally centred around 0). In each unit of each hidden layer, the weighted values are summed before going through a nonlinear activation function, responsible for introducing the nonlinearities in the model. Using a series of iterations known as epochs, the ANN will try to minimize a specific loss function (the mean squared error (MSE) in our case) comparing the processed values of the output layer with the ground truth ($y$). In order to avoid falling into local minima of the loss function, some regularisation is needed to prevent the ANN from overfitting (Hastie et al., 2009). To prevent overfitting during the training process (*i.e.*. to increase the ability of the model to generalize to new data), we used a classical regularization method called dropout, consisting in training iteratively smaller subparts of the ANN by randomly disconnecting a certain amount of connections between units. The introduction of Gaussian noise at the input of the ANN also helped to generalize, as it performs a similar effect to data augmentation. The main consequence of regularisation is generalization, for which the produced model is capable of better adapting to different configurations of the input data.

The hyperparameters used to configure the ANN are determined using cross-validation, in order to find the best performing combination of number of units, hidden layers, activation function, learning rate and regularisation method. Due to the relatively

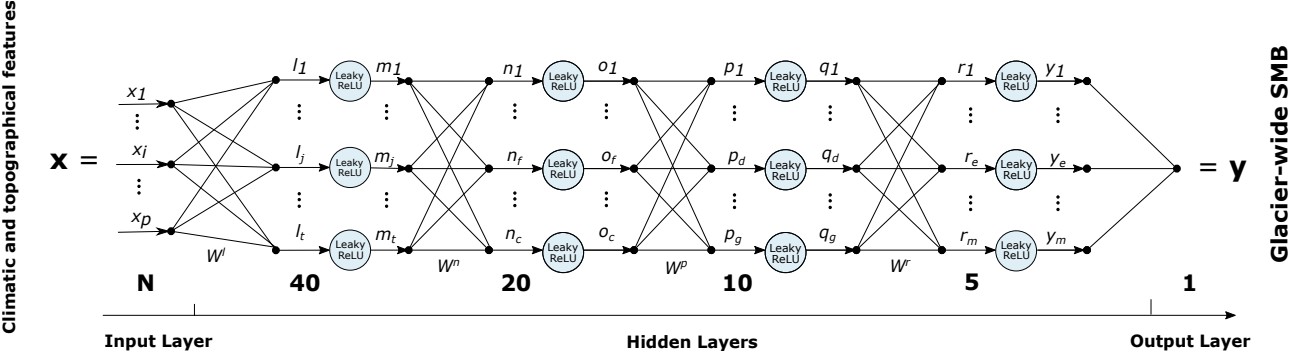

**Figure 3.** Deep Artificial Neural Network architecture used in ALPGM. The numbers indicate the number of neurons in each layer

small size of our dataset, we encountered the best performances with a quite small deep ANN, with a total of 6 layers (4 hidden layers) with a $(N, 40, 20, 10, 5, 1)$ architecture (Fig. 3), where $N$ is the number of selected features. Since the ANN already performs all the possible combinations between features (predictors), we use a reduced version of the training matrix from Eq. 1, with no combination of climatic and topographical features. Due to the relatively small size of the architecture, the best

dropout rates are small (Srivastava et al., 2014), and range between 0.3 and 0.01 depending on the number of units of each hidden layer. Leaky ReLUs have been chosen as the activation function, because of their widespread reliability and the fact they help prevent the "dead ReLU" problem, where certain neurons can stop "learning" (Xu et al., 2015). The He uniform initialization (He et al., 2015) has been used as it is shown to work well with Leaky ReLUs, and all unit bias were initialized to zero. In order to optimize the weights of the gradient descent, we used the RMSprop optimizer, for which we fine-tuned the

learning rate, obtaining the best results at 0.0005 in space and 0.02 in time. Each batch was normalized before applying the activation function in order to accelerate the training (Ioffe and Szegedy, 2015).

Like for many other geophysical processes found in nature, extreme annual glacier-wide SMB values occur much less often than average values, approximately following an unbounded Gumbel-type distribution (Thibert et al., 2018). From a statistical point of view, this means that ANN will "see" few extreme values and will accord less importance to them. For future

projections in a warmer climate, extreme positive glacier-wide SMB balances should not be the main concern of glacier models. However, extreme negative annual glacier-wide SMB values should likely increase in frequency, so it is in the modeller's interest to reproduce them as well as possible. Setting the sample weights as the inverse of the probability density function during the ANN training can partly compensate for the imbalance of a dataset. This boosts the performance of the model for the extreme values, at the cost of sacrificing some performance on more average values, which can be seen as a $r^2$/RMSE

trade-off (see Fig. 6 and 9 from the case study). The correct setting of the sample weights allows the modeller to adapt the ANN to each dataset and application.

## 2.3 Glacier geometry update

Since the first component of ALPGM simulates annual glacier-wide SMBs, these changes in mass need to be redistributed over the glacier surface-area in order to reproduce glacier dynamics. This redistribution is applied using the $\Delta$h parameterization. The idea was first developed by Jóhannesson et al. (1989) and then adapted and implemented by Huss et al. (2008). The main idea behind it is to use two or more DEMs covering the study area. These DEMs should have dates covering a period long enough (which will be later discussed in detail). By subtracting them, the changes in glacier surface elevation over time can be computed, which corresponds to a change in thickness (considering no basal erosion). Then, these thickness changes are normalized and considered as a function of the normalized glacier altitude. This $\Delta$h function is specific for each glacier and represents the normalized glacier thickness evolution over its altitudinal range. One advantage of such a parametrized approach is that it implicitly considers the ice flow which redistributes the mass from the accumulation to the ablation area. In order to make the glacier volume evolve in a mass-conserving fashion, we apply this function to the annual glacier-wide SMB values in order to scale and distribute its change in volume.

As discussed in Vincent et al. (2014), the time period between the two DEMs used to calibrate the method needs to be long enough to show important ice thickness differences. The criteria will of course depend on each glacier and each period, but it will always be related to the achievable signal-to-noise ratio. Vincent et al. (2014) concluded that for their study on the Mer de Glace glacier (28.8 $km^2$, mean altitude = 2868 m.a.s.l.) in the French Alps, the 2003-2008 period was too short, due to the delayed response of glacier geometry to a change in surface mass balance. Indeed, the results for that 5-year period diverged from the results from longer periods. Moreover, the period should be long enough to be representative of the glacier evolution, which will often encompass periods with strong ablation and others with no retreat or even with positive SMBs.

Therefore, by subtracting the two DEMs, the ice thickness difference is computed for each specific glacier. These values can then be classified by altitude, thus obtaining an average glacier thickness difference for each pixel altitude. As a change to previous studies (Vincent et al., 2014; Huss and Hock, 2015; Hanzer et al., 2018; Vincent et al., 2019), we no longer work with altitudinal transects, but with individual pixels. In order to filter noise and artefacts coming from the DEM raster files, different filters are applied to remove outliers and pixels with unrealistic values, namely at the border of glaciers or where the surface slopes are high (refer to Supplements for detailed information). Our methodology thus allows to better exploit the available spatial information based on its quality, and not on arbitrary location within transects.

## 3 Case study: French alpine glaciers

### 3.1 Data

All data used in this case study is based on the French Alps (Fig. 4), located in the westernmost part of the European Alps, between 5.08° and 7.67°E, and 44° and 46°13'N. This region is particularly suited for the validation of a glacier evolution model because of the wealth of available data. Moreover, ALPGM has been developed as part of a hydro-glaciological study to understand the impact of the retreat of French alpine glaciers in the Rhône river catchment (97,800 $km^2$).

### 3.1.1 Glacier-wide surface mass balance

An annual glacier-wide SMB dataset, reconstructed using remote sensing based on changes in glacier volume and the snow line altitude, is used (Rabatel et al., 2016). This dataset is constituted by annual glacier-wide SMB values for 30 glaciers in the French Alps (Fig. 4) for 31 years, between 1984-2014. The great variety in topographical characteristics of the glaciers included in the dataset, with a good coverage of the three main clusters or groups of glaciers in the French Alps (Fig. 4), makes them an ideal training dataset for the model. Each of the clusters represents a different setup of glaciers with different contrasting latitudes (Écrins and Mont-Blanc), longitudes (Écrins and Vanoise), glacier size (smaller glaciers in Écrins and Vanoise *vs* larger ones in Mont-Blanc) and climatic characteristics with a Mediterranean influence towards the south of the study region. For more details regarding this dataset refer to Rabatel et al. (2016). Data from the Mer de Glace, Saint-Sorlin, Sarennes and Argentière glaciers is also used, coming from field observations from the GLACIOCLIM observatory. For some of these glaciers, glacier-wide SMB values are available since 1949, although only values from 1959 onwards were used to match the meteorological reanalysis. This makes a total of 32 glaciers (Argentière and Saint-Sorlin glaciers belonging to the two datasets), representing 1048 annual glacier-wide SMB values (taking into account some gaps in the dataset).

### 3.1.2 Topographical glacier data and altimetry

The topographical data used for the training of the glacier-wide SMB machine learning models is taken from the multitemporal inventory of the French Alps glaciers (e.g., Gardent et al., 2014) partly available through the GLIMS Glacier Database (NSIDC, 2005). We worked with the 1967, 1985, 2003 and 2015 inventories (Gardent et al., 2014, with 2015 update). Between these dates, the topographical predictors are linearly interpolated. On the other hand, in the glacier evolution component of ALPGM (Fig.1, step 5), the topographical data is re-computed every year for each glacier from the evolving and annually updated glacier-specific ice thickness and DEM rasters (Sect. 3.1.3). Since these raster files are estimates for the year 2003 (Farinotti et al. (2019) for the ice thickness), the full glacier evolution simulations can start the earliest at this date. For the computation of the glacier-specific geometry update functions, two DEMs covering the whole French Alps have been used: (1) one from 2011 generated from SPOT5 stereo-pair images, acquired on 15 October 2011; and (2) a 1979 aerial photogrammetric DEM from the French National Geographic Institute (Institut Géographique National, IGN), processed from aerial photographs taken around 1979. Both DEMs have an accuracy between 1 and 4 meters (Rabatel et al., 2016), and their uncertainties are negligible compared to many other parameters in this study.

### 3.1.3 Glacier ice thickness

Glacier ice thickness data come from Farinotti et al. (2019), hereafter F19, based on the Randolph Glacier Inventory v6.0 (RGI, Consortium, 2017). The ice thickness values represent the latest consensus estimate, averaging an ensemble of different methods based on the principles of ice flow dynamics to invert the ice thickness from surface characteristics.

We also have ice thickness data acquired by diverse field methods (seismic, ground penetrating radar or hot water drilling, Rabatel et al., 2018) for four glaciers of the GLACIOCLIM observatory. We compared these in situ thickness data, with

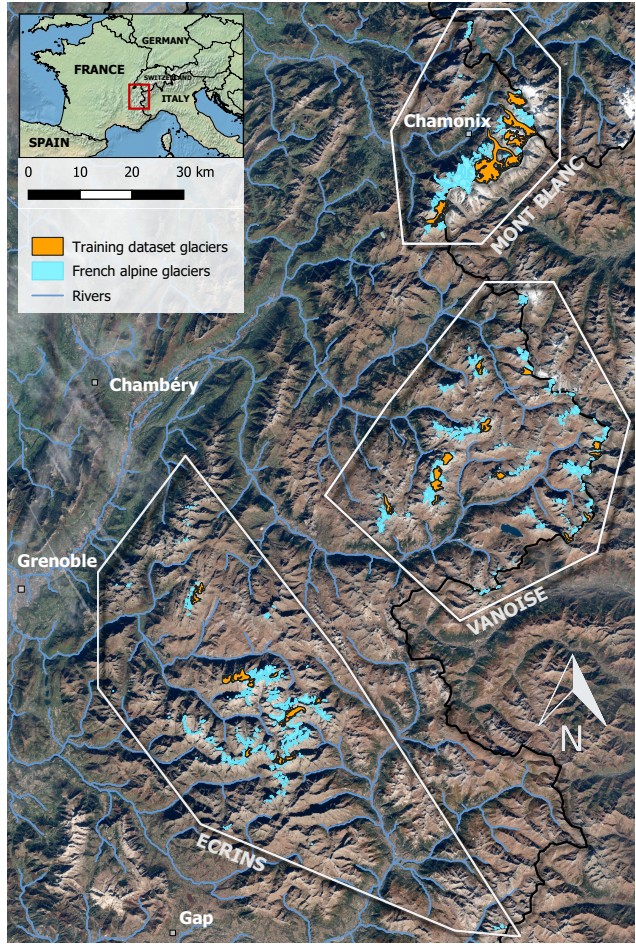

**Figure 4.** French alpine glaciers used for model training and validation and their classification into 3 clusters/regions (Écrins, Vanoise, Mont-Blanc). Coordinates of bottom left map corner: 44º32'N, 5º40'E, coordinates of the top right map corner: 46º08'N, 7º17'E.

the simulated ice thicknesses from F19 (refer to Supplements for detailed information). Although differences can be found (locally up to 100% in the worst cases), no systematic biases were found with respect to glacier local slope nor glacier altitude; therefore, no systematic correction was applied to the dataset. The simulated ice thicknesses for Saint-Sorlin (2 $km^2$, mean altitude = 2920 m.a.s.l., Écrins cluster) and Mer de Glace (28 $km^2$, mean altitude = 2890 m.a.s.l., Mont-Blanc cluster) glaciers

5 are satisfactorily modelled by F19. Mer de Glace's tongue presents local errors of about 50 m, peaking at 100 m (30% error) around 2000-2100 m.a.s.l, but the overall distribution of the ice is well represented. Saint Sorlin glacier follows a similar pattern, with maximum errors of around 20 m (20% error) at 2900 m.a.s.l. and a good representation of the ice distribution. The ice thicknesses for Argentière Glacier (12.8 $km^2$, mean altitude = 2808 m.a.s.l., Mont-Blanc cluster) and Glacier Blanc (4.7 $km^2$, mean altitude = 3196 m.a.s.l., Écrins cluster) are underestimated by F19 with an almost constant bias with respect to

10 altitude, as seen in Rabatel et al. (2018). Therefore, a manual correction was applied to the F19 datasets for these two glaciers

based on the field observations from the GLACIOCLIM observatory. A detailed plot (Fig. S2) presenting these results can be found in the supplementary material.

### 3.1.4 Climate data

In our French Alps case study, ALPGM is forced with daily mean near-surface (2 m) temperatures, daily cumulative snowfall and rain. The SAFRAN dataset is used to provide this data close to the glaciers' centroids. SAFRAN meteorological data (Durand et al., 2009) is a reanalysis of weather data including observations from different networks, and specific to the French mountain regions (Alps, Pyrenees and Corsica). Instead of being structured as a grid, data is provided at the scale of massifs, which are in turn divided into altitude bands of 300 meters and into 5 different aspects (north, south, east, west and flat).

## 3.2 Glacier-wide surface mass balance simulations: validation and results

In this section, we go through the selection of SMB predictors, we introduce the procedure for building machine learning SMB models, we assess their performance in space and time and we show some results of simulations using the French alpine glaciers dataset.

### 3.2.1 Selection of predictors

Statistical relationships between meteorological and topographical variables with respect to glacier-wide SMB are frequent in the literature for the European Alps (Hoinkes, 1968). Martin (1974) performed a sensitivity study on the SMB of the Saint-Sorlin and Sarennes glaciers (French Alps) with respect to multi-annual meteorological observations for the 1957-1972 period. Martin (1974) obtained a multiple linear regression function based on annual precipitation and summer temperatures, and he concluded that it could be further improved by differentiating winter and summer precipitations. Six and Vincent (2014) studied the sensitivity of the SMB to climate change in the French Alps from 1998 until 2014. They found that the variance of summer SMB is responsible for over 90% of the variance of the annual glacier-wide SMB. Rabatel et al. (2013, 2016) performed an extensive sensitivity analysis of different topographical variables (slope of the lowermost 20% of the glacier area, mean elevation, surface area, length, minimum elevation, maximum elevation, surface area change and length change) with respect to glacier ELA and annual glacier-wide SMBs of French alpine glaciers. Together with Huss (2012), who performed a similar study with SMB, the most significant statistical relationships were found for the lowermost 20% area slope, the mean elevation, glacier surface area, aspect and easting and northing. Rabatel et al. (2013) also determined that the climatic interannual variability is mainly responsible for driving the glacier equilibrium-line altitude temporal variability, whereas the topographical characteristics are responsible for the spatial variations in the mean ELA.

Summer ablation is often accounted for by means of cumulative positive degree days (CPDD). However, in the vast majority of studies, accumulation and ablation periods are defined between fixed dates (*e.g.*, 1st October - 30th April for the accumulation period in the northern mid-latitudes) based on optimizations. As discussed in Zekollari and Huybrechts (2018), these fixed periods may not be the best to describe SMB variability through statistical correlation. Moreover, the ablation season will likely

evolve in the coming century, due to climate warming. In order to overcome these limitations, we dynamically calculate each year the transition between accumulation and ablation seasons (and vice-versa) based on a chosen quantile in the CPDD (Fig. S3). We found higher correlations between annual SMB and ablation-period CPDD calculated using this dynamical ablation season. On the other hand, it was not the case for the separation between summer and winter snowfall. Therefore, we decided

5   to keep constant periods to account for winter (1st October-1st May) and summer (1st May-1st October) snowfalls, and to keep them dynamical for the CPDD calculation.

Following this literature review, vectors $\hat{\Omega}$ and $\hat{C}$ from (Eq. 1) read as:

$$\hat{\Omega} = \left[ \begin{array}{ccccccc} \overline{Z} & Z_{\mathrm{max}} & \alpha_{20\%} & \mathrm{Area} & \mathrm{Lat} & \mathrm{Lon} & \Phi \end{array} \right] \tag{2}$$

$$\hat{C} = \left[ \begin{array}{ccccc} \Delta CPDD & \Delta WS & \Delta SS & \Delta\overline{T}_{\mathrm{mon}} & \Delta\overline{S}_{\mathrm{mon}} \end{array} \right] \tag{3}$$

Where:

$\overline{Z}$: Mean glacier altitude

$Z_{\mathrm{max}}$: Maximum glacier altitude

$\alpha_{20\%}$: Slope of the lowermost 20% glacier altitudinal range

15   $Area$: Glacier surface area

$Lat$: Glacier latitude

$Lon$: Glacier longitude

$\Phi$: Cosine of the glacier's aspect (North = 0º)

$\Delta CPDD$: CPDD (Cumulative Positive Degree Days) anomaly

20   $\Delta WS$: Winter snow anomaly

$\Delta SS$: Summer snow anomaly

$\Delta\overline{T}_{\mathrm{mon}}$: Average temperature anomaly for each month for the hydrological year

$\Delta\overline{S}_{\mathrm{mon}}$: Average snowfall anomaly for each month for the hydrological year

For the linear machine learning models training, we chose a function $f$ that linearly combines $\hat{\Omega}$ and $\hat{C}$, generating new combined predictors (Eq. 4). In $\hat{C}$, only $\Delta CPDD$, $\Delta WS$, and $\Delta SS$ are combined, to avoid generating an unnecessary amount of predictors with the combination of $\hat{\Omega}$ with $\Delta\overline{T}_{\mathrm{mon}}$ and $\Delta\overline{S}_{\mathrm{mon}}$.

$$
\begin{aligned}
SMB_{g,y} = (&(a_1\overline{Z} + a_2 Z_{\max} + a_3\alpha_{20\%} + a_4 Area + a_5 Lat + a_6 Lon + a_7\Phi + a_8)\Delta CPDD + \\
&(b_1\overline{Z} + b_2 Z_{\max} + b_3\alpha_{20\%} + b_4 Area + b_5 Lat + b_6 Lon + b_7\Phi + b_8)\Delta SS + \\
&(c_1\overline{Z} + c_2 Z_{\max} + c_3\alpha_{20\%} + c_4 Area + c_5 Lat + c_6 Lon + c_7\Phi + c_8)\Delta WS + \\
&d_1\overline{Z} + d_2 Z_{\max} + d_3\alpha_{20\%} + d_4 Area + d_5 Lat + d_6 Lon + d_7\Phi + d_8 + d_n\Delta\overline{T}_{\mathrm{mon}} + d_m\Delta\overline{S}_{\mathrm{mon}} + \varepsilon)_{g,y}
\end{aligned}
\tag{4}
$$

32 glaciers over variable periods between 31 and 57 years result in 1048 glacier-wide SMB ground truth values. For each glacier-wide SMB value, 55 predictors were produced following Eq. 4: 33 combined predictors, with $\Delta\overline{T}_{\mathrm{mon}}$ and $\Delta\overline{S}_{\mathrm{mon}}$ accounting for 12 predictors each, one for each month of the year. All these values combined produce a 1048x55 matrix, given as input data to the OLS and Lasso machine learning libraries. Early Lasso tests (not shown here) using only the predictors from Eq. 2 and 3 demonstrated the benefits of expanding the number of predictors, as it is later shown in Fig. 5. For the training of the ANN, no combination of topo-climatic predictors is done as previously mentioned (Sect. 2.2.4), since it is already done internally by the ANN.

### 3.2.2 Causal analysis

By running the Lasso algorithm on the dataset based on Eq. 2 and 3, we obtain the contribution of each predictor in order to explain the annual glacier-wide SMB variance. Regarding the climatic variables, accumulation-related predictors (winter snowfall, summer snowfall as well as several winter, spring and even summer months), appear as the most important predictors. Ablation-related predictors also seem to be relevant, mainly with CPDD and summer and shoulder season months (Fig. 5). Interestingly, meteorological conditions in the transition months are crucial for the annual glacier-wide SMB in the French Alps: (1) October temperature is determinant for the transition between the ablation and the accumulation season, favouring a lengthening of melting when temperature remains positive, or conversely allowing snowfalls that protect the ice and contribute to the accumulation when temperatures are negative; (2) March snowfall has a similar effect: positive anomalies contribute to the total accumulation at the glacier surface, and a thicker snow pack will delay the snow/ice transition during the ablation season leading to a less negative ablation rate (e.g. Fig. 6b, Réveillet et al., 2018). Therefore, meteorological conditions of these transition months seem to strongly impact the annual glacier-wide SMB variability, since their variability oscillates between positive and negative values, unlike the months in the heart of summer or winter.

In a second term, topographical predictors do play a role, albeit a secondary one. The slope of the 20% lowermost altitudinal range, the glacier area, the glacier mean altitude and aspect help to modulate the glacier-wide SMB signal, which unlike point or altitude-dependent SMB, partially depends on glacier topography (Huss et al., 2012). Moreover, latitude and longitude are among the most relevant topographical predictors, which for this case study are likely to be used as bias correctors of

precipitation of the SAFRAN climate reanalysis. SAFRAN is suspected of having a precipitation bias, with higher uncertainties for high altitude precipitations (Vionnet et al., 2016). Since the French Alps present an altitudinal gradient, with higher altitudes towards the eastern and the northern massifs, we found that the coefficients linked to latitude and longitude enhanced glacier-wide SMBs with a north-east gradient.

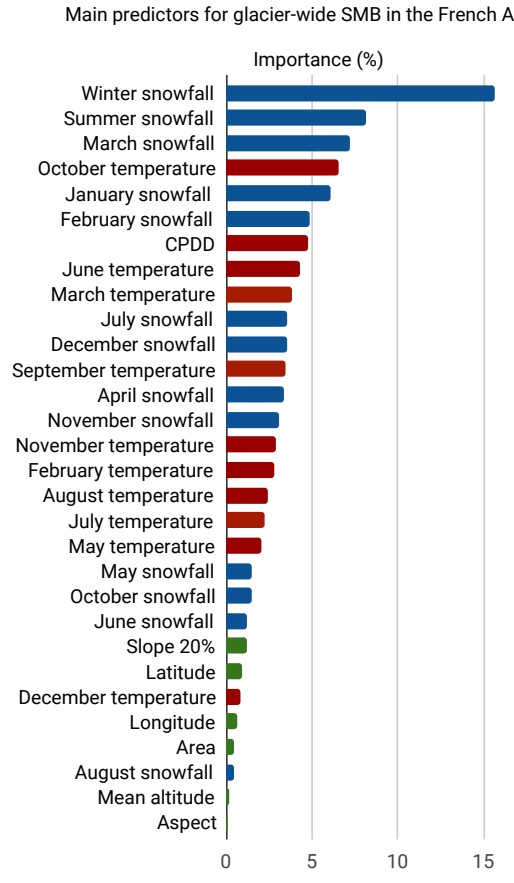

**Figure 5.** Contribution to the total variance of the 30 top topo-climatic predictors out of 55 predictors using Lasso. Green bars indicate predictors including topographical features, blue ones including accumulation-related features, and red ones including ablation-related features

### 3.2.3 Spatial predictive analysis

In order to evaluate the performance of the machine learning SMB models in space, we perform a leave-one-glacier-out (LOGO) cross-validation. For relatively small datasets like the one used in this study, cross-validation ensures that the model is validated on the full dataset. Such validation aims at understanding the model's performance for predictions on other glaciers for the same time period as during the training.

An important aspect is the comparison between linear and nonlinear machine learning algorithms used in this study. Steiner et al. (2005) already proved that a nonlinear ANN improved the results with respect a classic stepwise multiple linear regression. Here, we draw a similar comparison using more advanced methods for a larger dataset: OLS and Lasso as linear machine learning algorithms and a deep ANN as a nonlinear one. We observed significant differences between OLS, Lasso and deep learning, both in terms of explained variance ($r^2$) and accuracy (RMSE) of predicted glacier-wide SMBs. On average, we found improvements between +55% and +61% in the explained variance (from 0.49 to 0.76-0.79) using the nonlinear deep ANN compared to Lasso, whereas the accuracy was improved up to 45% (from 0.74 to 0.51-0.62). This means that 27% more variance is explained with a nonlinear model in the spatial dimension for glacier-wide SMB in this region. See Fig. 6 for a full summary of the results. An interesting consequence of the nonlinearity of the ANN is the fact that it better captures extreme SMB values compared to a linear model. A linear model can correctly approximate the main cluster of values around the median, but the linear approximation performs poorly for extreme annual glacier-wide SMB values. The ANN solves this problem, with an increased explained variance which translates into a better accuracy for extreme SMB values, even without the use of sample weights (Fig. 6).

As a consequence, the added value of deep learning is especially relevant on glaciers with steeper annual changes in glacier-wide SMB (Fig. 7a). The use of sample weights can scale up or down this factor, thus playing with a performance trade-off depending on how much one wants to improve the model's behaviour for extreme SMB values.

Overall, deep learning results in a lower error throughout all the glaciers in the dataset when evaluated using LOGO cross-validation (Fig. 8). Moreover, the bias is also systematically reduced, but it is strongly correlated to the one from Lasso.

### 3.2.4 Temporal predictive analysis

In order to evaluate the performance of the machine learning SMB models in time, we perform a leave-one-year-out (LOYO) cross-validation. This validation serves to understand the model's performance for past or future periods outside the training time period. The best results achieved for Lasso make no use of any monthly average temperature or snowfall, suggesting that these features are not relevant for temporal predictions unlike the spatial case.

As in Sect. 3.2.3, the results between the linear and nonlinear machine learning algorithms were compared. Interestingly, using LOYO, the differences between the different models were even greater than for spatial validation, revealing the more complex nature of the information in the temporal dimension. As illustrated by Fig. 9, we found remarkable improvements between the linear Lasso and the nonlinear deep learning in both the explained variance (between +94% and +108%) and accuracy (between +32% and +58%). This implies that 35% more variance is explained using a nonlinear model in the temporal dimension for glacier-wide SMB balance in this region. Deep learning manages to keep very similar performances between the spatial and temporal dimensions, whereas the linear methods see their performance affected most likely due to the increased nonlinearity of the SMB reaction to meteorological conditions.

A more detailed year by year analysis reveals interesting information about the glacier-wide SMB data structure. As seen in Fig. 10, the years with the worst deep learning precision are 1984, 1985 and 1990. All these three hydrological years present a high spatial variability in observed (or remotely-sensed) SMBs: very positive SMB values in general for 1984 and 1985

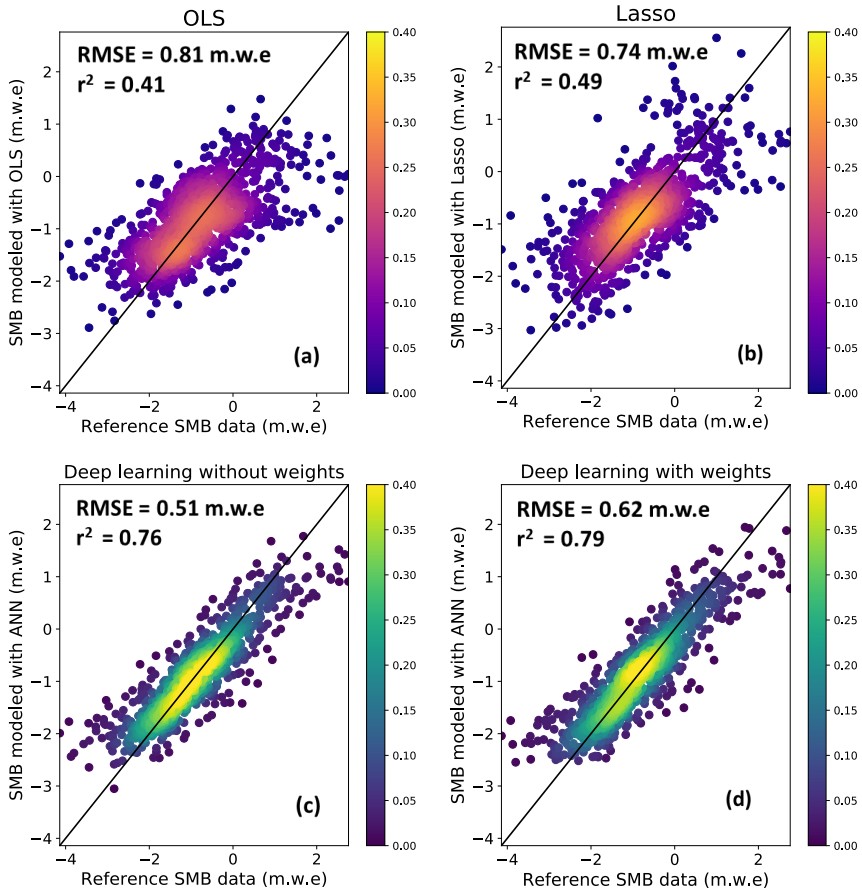

**Figure 6.** Evaluation of modelled annual glacier-wide SMB against the ground truth SMB data (both in m.w.e. $a^{-1}$) using Leave-One-Glacier-Out cross-validation. The colour (purple-orange for linear; blue-green for nonlinear) indicates frequency based on the probability density function. The black line indicates the reference one-to-one line. a) Scatter plot of the OLS model results; b) Scatter plot of the Lasso linear model results; Scatter plots of the deep artificial neural network nonlinear models without (c) and with sample weights (d)

with few slightly negative values, and extremely negative SMB values in general for 1990 with few almost neutral values. These complex configurations are clearly outliers within the dataset, which push the limits of the nonlinear patterns found by the ANN. The situation becomes even more evident with Lasso, which struggles to resolve these complex patterns and often performs poorly where the ANN succeeds (*e.g.*, years 1996, 2012 or 2014). The important bias present only with Lasso is

5 representative of its lack of complexity towards nonlinear structures, which results in an underfitting of the data. The average error is not bad, but it shows a high negative bias for the first half of the period, which mostly has slightly negative glacier-wide SMBs, and a high positive bias for the second half of the period, which mostly has very negative glacier-wide SMB values.

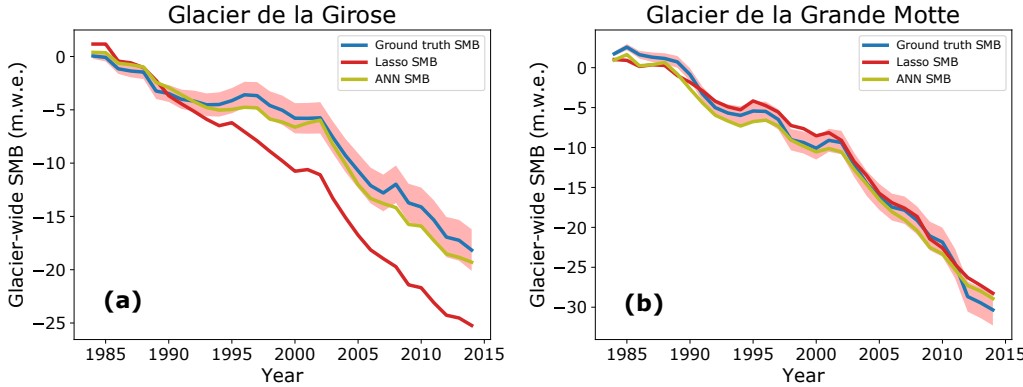

**Figure 7.** Examples of cumulative glacier-wide SMB (m.w.e.) simulations against the ground truth SMB data. The pink envelope indicates the accumulated uncertainties from the ground truth data. The deep learning SMB model has not been trained with sample weights in these illustrations.

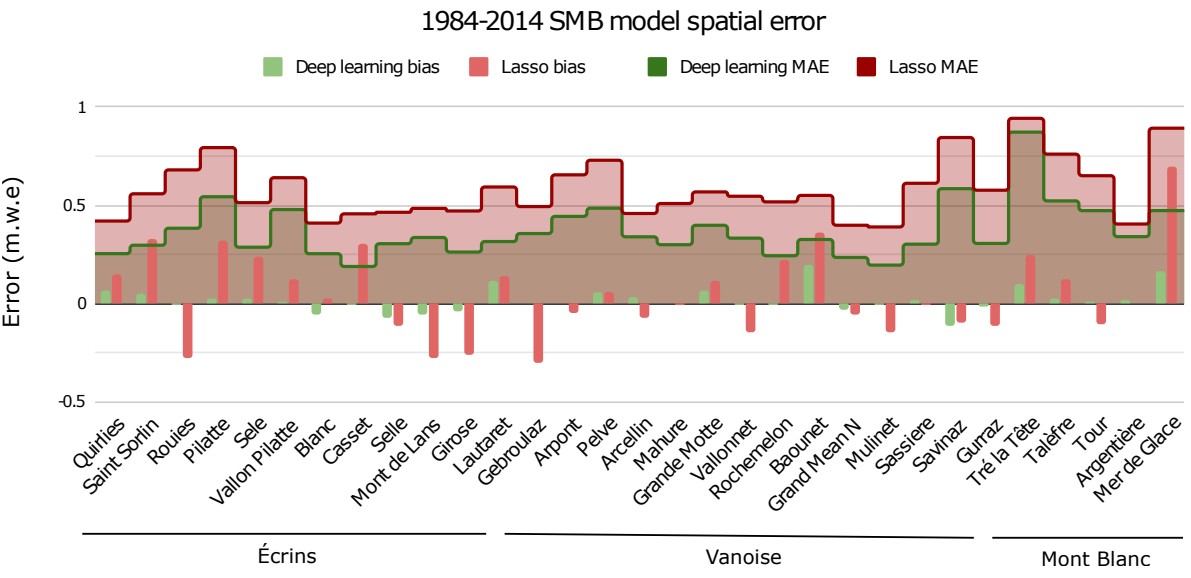

**Figure 8.** Mean average error (MAE) and bias (vertical bars) for each glacier of the training dataset structured by clusters for the 1984-2014 LOGO glacier-wide SMB simulation. No clear regional error patterns arise

### 3.2.5 Spatiotemporal predictive analysis

Once the specific performances in the spatial and temporal dimensions have been assessed, the performance in both dimensions at the same time is evaluated using Leave-Some-Years-and-Glaciers-Out (LSYGO) cross-validation. 64 folds were built, with test folds being comprised of data for 2 random glaciers on 2 random years, and train folds of all the data except the 2 years

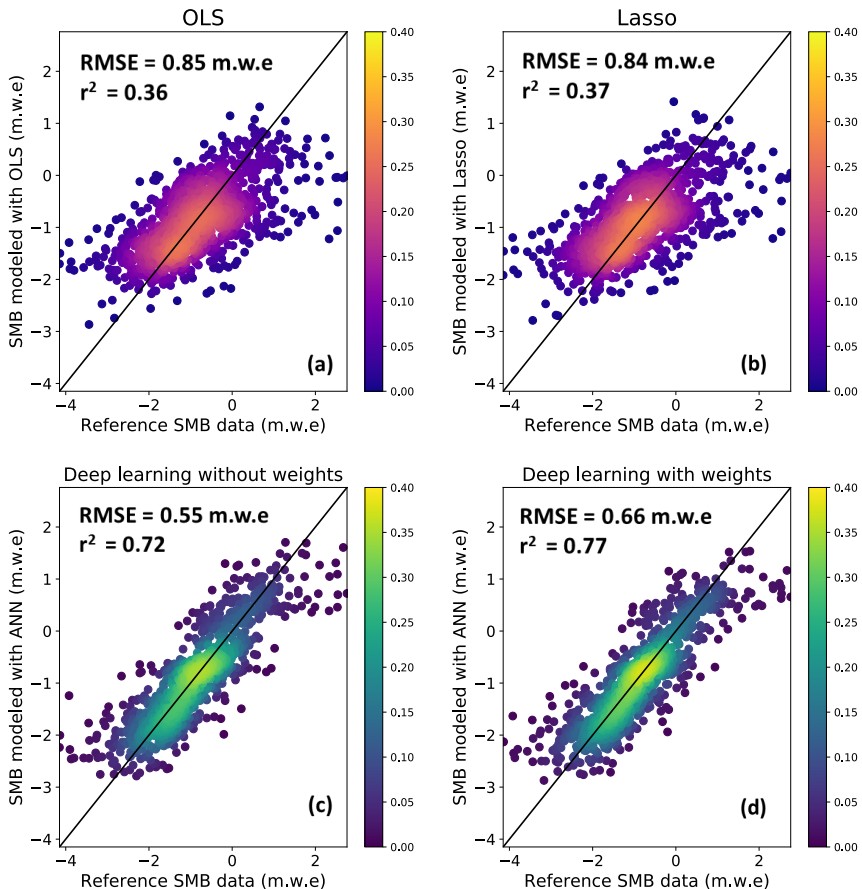

**Figure 9.** Evaluation of modelled annual glacier-wide SMB against the ground truth SMB data (both in m.w.e. $a^{-1}$) using Leave-One-Year-Out cross-validation. The colour (purple-orange for linear; blue-green for nonlinear) indicates frequency based on the probability density function. The black line indicates the reference one-to-one line. a) Scatter plot of the OLS model results; b) Scatter plot of the Lasso linear model results; Scatter plots of the deep artificial neural network nonlinear models without (c) and with sample weights (d).

(for all glaciers) and the 2 glaciers (for all years) present in the test fold. These combinations are quite strict, implying that for every 4 tested values we need to drop between 123 and 126 values for training, depending on the glacier and year, to respect the spatiotemporal independence (Roberts et al., 2017).

The performance of LSYGO is similar to LOYO, with a RMSE of 0.51 m.w.e. and a coefficient of determination of 0.77

5 (Fig. S5). This is reflected in the fact that very similar ANN hyperparameters were used for the training. This means that the deep learning SMB model is successful in generalizing and it does not overfit the training data.

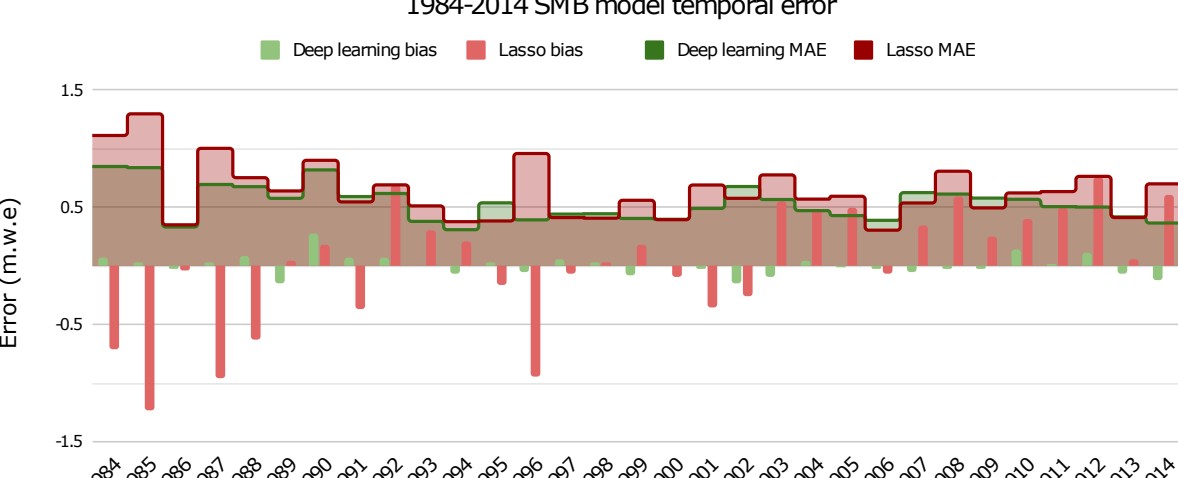

**Figure 10.** Mean average error (MAE) and bias (vertical bars) for each year of the training dataset for the 1984-2014 LOYO glacier-wide SMB simulation.

## 3.3 Glacier geometry evolution: Validation and results

As mentioned in Sect. 2.3, the $\Delta$h parameterization has been widely used in many studies (e.g., Huss et al., 2008, 2010; Vincent et al., 2014; Huss and Hock, 2015, 2018; Hanzer et al., 2018; Vincent et al., 2019). It is not in the scope of this study to evaluate the performance of this method, but we present the approach developed in ALPGM to compute the $\Delta$h functions and
5   show some examples for single glaciers to illustrate how these glacier-specific functions perform compared to observations. For the studied French alpine glaciers, the 1979-2011 period is used. This period was proved by Vincent et al. (2014) to be representative of Mer de Glace's secular trend. Other sub-periods could have been used, but it was shown that they did not necessarily improve the performance. In addition, the 1979 and 2011 DEMs are the only ones available that cover all the French alpine glaciers. Within this period, some years with neutral to even positive surface mass balances in the late 1970s and
10   early 1980s can be found, as well as a remarkable change from 2003 onward with strongly negative surface mass balances, following the heatwave that severely affected the western Alps in summer 2003.

   The glacier-specific $\Delta$h functions are computed for glaciers $\geq 0.5\ km^2$, which represented about 80% of the whole glacia-rized surface of the French Alps in 2015 (some examples are illustrated in the Supplement Fig. S4). For the rest of very small glaciers ($< 0.5\ km^2$), a standardized flat function is used in order to make them shrink equally at all altitudes. This is done to
15   simulate the fact that generally, the equilibrium line of very small glaciers has surpassed the glacier's maximum altitude, thus shrinking from all directions and altitudes in summer. Moreover, due to their reduced size and altitudinal range, the ice flow no longer has the same importance as for larger or medium sized glaciers.

In order to evaluate the performance of the parameterized glacier dynamics of ALPGM, coupled with the glacier-wide SMB component, we compared the simulated glacier area of the 32 studied glaciers with the observed area in 2015 from the most up-to-date glacier inventory in the French Alps. Simulations were started in 2003, for which we used the F19 ice thickness dataset. In order to take into account the ice thickness uncertainties, we ran three simulations with different versions of the initial ice

5     thickness: the original data, -30% and +30% of the original ice thickness in agreement with the uncertainty estimated by the authors. Moreover, in order to take into account the uncertainties in the Δh glacier geometry update function computation, we added a ±10% variation in the parameterized functions (Fig. 11).

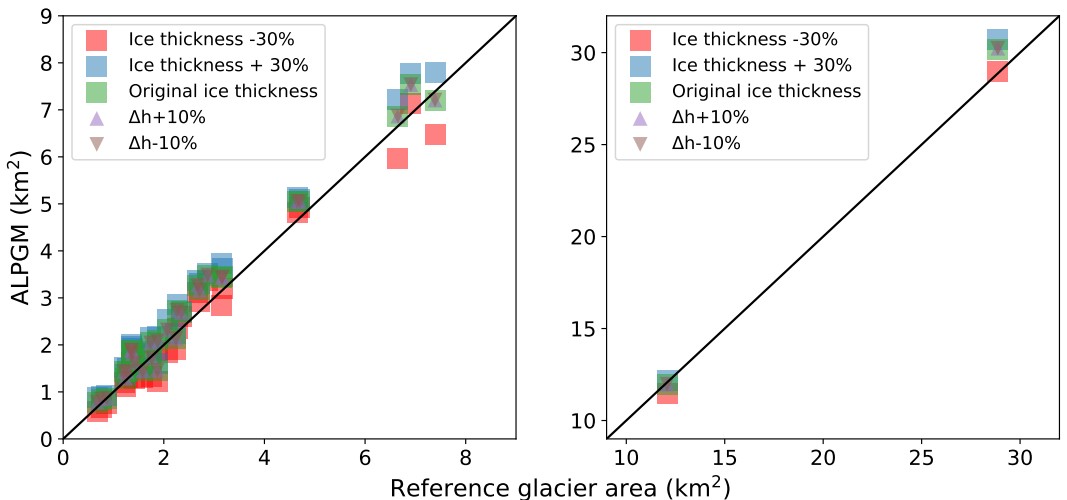

**Figure 11.** Simulated glacier areas for the 2003-2015 period for the 32 study glaciers using a deep learning SMB model without weights. Squares indicate the different F19 initial ice thicknesses used taking into account their uncertainties and triangles indicate the uncertainties linked to the glacier-specific geometry update functions. For better visualisation, the figure is split in two with the two largest French glaciers on the right.

Overall, the results illustrated in Fig. 11 show a good agreement with the observations. Even for a 12-year period, the initial ice thickness remains the largest uncertainty, with almost all glaciers falling within the observed area when taking it

10     into account. The mean error in simulated surface area was of 10.7% with the original F19 ice thickness dataset. Other studies using the Δh parameterization already proved that the initial ice thickness is the most important uncertainty in glacier evolution simulations, together with the choice of a GCM for future projections (Huss and Hock, 2015).

## 4 Discussion and perspectives

### 4.1 Linear methods still matter

Despite the fact that deep learning often outperforms linear machine learning and statistical methods, there is still a place for such methods in modelling. Indeed, unlike ANNs, simpler regularised linear models such as Lasso allow an easy interpretation of the coefficients associated to each input feature, which helps to understand the contribution of each of the chosen variables to the model. This means that linear machine learning methods can be used for both prediction and causal analysis. Training a linear model in parallel to an ANN has therefore the advantage to provide a simpler linear alternative which can be used to understand the dataset. Moreover, seeing the contribution of each coefficient, one can reduce the complexity of the dataset by keeping only the most significant predictors. Finally, a linear model serves as well as a reference to highlight and quantify the nonlinear gains obtained by deep learning.

### 4.2 Training deep learning models with spatiotemporal data

The creation and training of a deep ANN requires a certain knowledge and strategy with respect to the data and study focus. When working with spatiotemporal data, the separation between training and validation becomes tricky. The spatial and temporal dimensions in the dataset cannot be ignored, and strongly affect the independence between training and validation data (Roberts et al., 2017; Oliveira et al., 2019). Depending on how the cross-validation is performed, the obtained performance will be indicative of one of these two dimensions. As it is shown in Sect. 3.2.3, the ANNs and especially the linear modelling approaches had more success in predicting SMB values in space than in time. This is mostly due to the fact that the glacier-wide SMB signal has a greater variability and nonlinearities in time than in space, with climate being the main driver of the annual fluctuations in SMB, whereas geography, and in particular the local topography, modulates the signal between glaciers (Huss, 2012; Rabatel et al., 2016; Vincent et al., 2017). Consequently, linear models find it easier to make predictions on a given period of time for other glaciers elsewhere in space, than for time periods outside the training. Nonetheless, the deep learning SMB models were capable of equally capturing the complex nonlinear patterns in both the spatial and temporal dimensions.

In order to cope with the specific challenges related to each type of cross-validation, there are several hyperparameters that can be modified to adapt the ANN's behaviour. Due to the long list of hyperparameters intervening in an ANN, it is not advisable to select them using brute force with a grid search or cross-validation. Instead, initial tests are performed in a subset of random folds to narrow down the range of best performing values, before moving to the full final cross-validations for the final hyperparameter selection. Moreover, the ANN architecture plays an important role: the number of neurons as well as the number of hidden layers will determine the ANN's complexity and its capabilities to capture hidden patterns in the data. But the larger the architecture, the higher are the chances to overfit the data. This undesired effect can be counterbalanced using regularization. The amount of regularization (dropout and Gaussian noise in our case, see Sect. 2.2.4) used in the training of the ANN necessarily introduces some trade-offs. The greater the dropout, the more we will constrain the learning of the ANN so the higher the generalization will be, until a certain point, where relevant information will start to be lost and performance will drop. On the other hand, the learning rate to compute the stochastic gradient descent, which tries to minimize the loss function,

also plays an important role: smaller learning rates generally result in a slower convergence towards the absolute minima, thus producing models with better generalization. By balancing all these different effects, one can achieve the accuracy versus generalization ratio that best suits a certain dataset and model in terms of performance. Nonetheless, one key aspect in machine learning models is data: expanding the training dataset in the future will allow to increase the complexity of the model and its

performance. Consequently, machine learning models see their performance improved as time goes by, with new data becoming available for training.

Although the features used as input for the model are classical descriptors of the topographical and meteorological conditions of the glaciers, it is worth mentioning that applying the model in different areas or with different data sources would likely require a re-training of the model due to possible biases: different regions on the globe may have other descriptors of importance

but also different measuring techniques will likely have different biases.

## 4.3    Perspectives on future applications of deep learning in glaciology

The currently used meteorological variables in the deep ANN of ALPGM's SMB component are based on the classic degree-day approach, which relies only on temperature and precipitation. However, the model could be trained with variables involved in more complex models, such as SEB-type models, for which the longwave and shortwave radiation, as well as the turbulent

fluxes and albedo intervene. The current model framework allows flexibility in the choice and number of input variables that can reflect different degrees of complexity for the resolved processes. Despite the fact that it has been shown that for glaciers in the European Alps there is almost no added value in transitioning from a simple degree-day to a SEB model for annual glacier-wide SMB simulations (e.g., Réveillet et al., 2017), it could be an interesting way to expand the training dataset for glaciers in tropical and subtropical regions, where shortwave radiation plays a much more important role (Benn and Evans,

2014). Maussion et al. (2015) followed a similar approach with linear machine learning in order to calibrate a regression-based downscaling model that linked local SEB/SMB fluxes to atmospheric reanalysis variables.

In this work, we also evaluated the resilience of the deep learning approach: since many glacierized regions in the world do not have the same amount of data used in this study, we trained an ANN only with monthly average temperature and snowfall, without any topographical predictors, to see until which point the algorithm is capable of learning from minimal data. The

results were quite interesting, with a coefficient of determination of 0.68 (against 0.76 from the full model) and a RMSE of 0.59 m.w.e. $a^{-1}$ (against 0.51 from the full model). These results indicate that meteorological data is the primary source of information, determining the interannual high frequency variability of the glacier-wide SMB signal. On the other hand, the "bonus" of topographical data helps to modulate the high frequency climate signal, by adding a low frequency component to better differentiate glaciers and the topographical characteristics included in the glacier-wide SMB data (Huss et al., 2012).

The fact that glacier-wide SMB is influenced by glacier topography poses the question of determining if the simulated glacier geometries can correctly reproduce topographical observations, needed to represent the topographical feedback present in glacier-wide SMB signals. These aspects are analyzed and discussed in Sect. 3 of the Supplementary material, showing small differences between the observed and simulated topographical parameters for the 2003-2015 period (Table S1). Additionally, the simulated glacier-wide SMBs using simulated topographical parameters show very small differences (0.069 m.w.e. $a^{-1}$

on average) compared to simulations using topographical observations (Fig. S6). Since glacier ice thickness estimates date from the year 2003 (Farinotti et al., 2019), our validation period can only encompass 12 years. According to all the available data for validation, our model seems to be able to correctly reproduce the glacier geometry evolution, but since the 2003-2015 validation period is quite short, the validation performance might not be representative when dealing with future glacier evolu-

tion projections of several decades. Consequently, these aspects will have to be taken into account for future studies using this modelling approach for projections. Moreover, the cross-validation results of the SMB model(s) (Fig. 6-10) are representative of the performance of predictions using topographical observations. Despite the small differences found between simulated and observed topographical parameters, the SMB model's performance might be slightly different than the performance found in the cross-validation analysis. Therefore, it would be interesting for future studies to investigate the use of point SMB data,

which could avoid the complexities related to the influence of glacier topography in glacier-wide SMB.

A nonlinear deep learning SMB component like the one used for ALPGM could provide an interesting alternative to classical SMB models used for regional modelling. The comparison with other SMB models is beyond the scope of this study, but it would be worth investigating to quantify the specific gains that could be achieved by switching to a deep learning modelling approach. Nonetheless, the linear machine learning models trained with the CPDD and cumulative snowfall used in this study

behave in a similar way to a calibrated temperature-index model. Even so, we believe that future efforts should be taken towards physics-informed data science glacier SMB and evolution modelling. Adding physical constraints in ANNs, with the use of physics-based loss functions and/or architectures (e.g., Karpatne et al., 2018), would allow to improve our understanding and confidence in predictions, reduce our dependency on big datasets, and to start bridging the gap between data science and physical methods (Karpatne et al., 2017; de Bezenac et al., 2018; Lguensat et al., 2019; Rackauckas et al., 2020). Deep learning

can be of special interest once applied in the reconstruction of SMB time series. More and more SMB data is becoming available thanks to the advances in remote sensing (e.g., Brun et al., 2017; Zemp et al., 2019; Dussaillant et al., 2019), but these datasets often cover limited areas and the most recent time period in the studied regions. An interesting way of expanding a dataset would be to use a deep learning approach to fill the data gaps, based on the relationships found in a subset of glaciers as in the case study presented here. Past SMB time series of vast glaciarized regions could thereby be reconstructed, with potential

applications in remote glaciarized regions such as the Andes or High Mountain Asia.

## 5  Conclusions

We presented a novel approach to simulate and reconstruct glacier-wide SMB series using deep learning for individual glaciers at a regional scale. This method has been included as a SMB component in ALPGM (Bolibar, 2019), a parameterized regional glacier evolution model, following an alternative approach to most physical and process-based glacier models. The data-driven

glacier-wide SMB modelling component is coupled with a glacier geometry update component, based on glacier-specific parameterized functions. Deep learning is shown to outperform linear methods for the simulation of glacier-wide SMB with a case study of French alpine glaciers. By means of cross-validation, we demonstrated how important nonlinear structures (up to 35%) coming from the glacier and climate systems in both the spatial and temporal dimensions are captured by the deep

ANN. Taking into account this nonlinearity substantially improved the explained variance and accuracy compared to linear statistical models, especially in the more complex temporal dimension. As we have shown in our case study, deep ANNs are capable of dealing with relatively small datasets, and they present a wide range of configurations to generalize and prevent overfitting. Machine learning models benefit from the increasing number of available data, which makes their performance constantly improve as time goes by.

Deep learning should be seen as an opportunity by the glaciology community. Its good performance for SMB modelling in both the spatial and temporal dimensions shows how relevant it can be for a broad range of applications. Combined with in situ or remote sensing SMB estimations, it can serve to reconstruct SMB time series for regions or glaciers with already available data for past and future periods, with potential applications in remote regions such as the Andes or the high mountains of Asia. Moreover, deep learning can be used as an alternative to classical SMB models as it is done in ALPGM: important nonlinearities from the glacier and climate systems are potentially ignored by these mostly linear models, which could give an advantage to deep learning models in regional studies. It might still be too early for the development of such models in certain regions which lack consistent datasets with a good spatial and temporal coverage. Nevertheless, upcoming methods adding physical knowledge to constrain neural networks (e.g., Karpatne et al., 2018; Rackauckas et al., 2020) could provide interesting solutions to the limitations of our current method. By incorporating prior physical knowledge in neural networks, the dependency on big datasets would be reduced, and it would allow to transition towards more interpretable physics-informed data science models.

*Code and data availability.* The source code of ALPGM is available at https://github.com/JordiBolibar/ALPGM with its DOI (10.5281/zenodo.3609136) for the v1.1 release. All scripts used to generate plots and results are included in the repository. The detailed information regarding glaciers used in the case study and the SMB model performance per glacier are included in the Supplementary material.

*Author contributions.* J. Bolibar developed ALPGM, analysed the results and wrote the paper. A. Rabatel did the glaciological analysis, contributed with the remote sensing SMB data, and conceived the study together with I. Gouttevin, T. Condom and J. Bolibar. C. Galiez and J. Bolibar developed the deep and machine learning modelling approach, and E. Sauquet contributed in the statistical analysis. All authors discussed the results and helped to develop the paper.

*Competing interests.* The authors declare that they do not have any competing interests.

*Acknowledgements.* This study has been possible through funding from the ANR VIP_Mont-Blanc (ANR-14 CE03-0006-03), the BERGER project (Pack Ambition Recherche funded by the AuRA region), the LabEx OSUG@2020 (Investissements d'avenir - ANR10 LABX56) and the CNES, via the KALEIDOS-Alpes and ISIS projects. Data has been provided by Météo-France (SAFRAN dataset) and the GLACIOCLIM

National Observation Service (*in situ* glaciological data). The authors would like to thank Eduardo Pérez-Pellitero (Max Planck Institute for Intelligent Systems) for the interesting discussions regarding deep learning, Benjamin Renard (Irstea Lyon) for his insightful comments on statistics, as well as Fabien Maussion and an anonymous reviewer for their constructive review comments which have helped to improve the overall quality and clarity of the manuscript.

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
