# Peer review of "Deep learning applied to glacier evolution modelling"

_The Cryosphere, 2019_

## Referee Comment (RC1) · Anonymous Referee #1 · 28 Aug 2019

Deep learning applied to glacier evolution modelling: Manuscript review

The research presented in this manuscript shows promising results in the application of an ANN model used for surface mass balance modelling. The manuscript is, for the most part, well organized. The manuscript can be greatly improved by increasing clarity and specificity throughout. I hope that my comments are helpful to the authors in this effort. No single one of my comments identifies a major flaw with the manuscript; rather, there are many small changes that I believe can be made to improve the quality of the paper. I have organized my comments in sequential order by section, preceded by one general note.

General Note:

The difference between "machine learning" and "deep learning" is not clearly defined in the literature, but a 6-layer ANN is likely at the very tip of what may constitute "deep Printer-friendly version

learning". Considering that, as you note, deep learning is not a common tool among the glaciological community, it would be good to provide further context as to what an ANN is (a type of model, which is relatively simple in the deep learning world as compared to, say, a convolutional neural network or long short-term memory network) versus what deep learning is. I believe that this is required especially because the title refers to deep learning broadly, not a deep ANN specifically, and it should be made clear that there is much more to deep learning than ANNs.

Sequential Notes:

Page 1, Line 22: What does "individual glaciers at regional scale" mean? Do you mean to say, "individual glaciers within the same region"?

1 Introduction: Page 1, Line 25: "... being climate proxies which can clearly depict the evolution of climate for the global audience"; remove "clearly"—if the evolution of climate was clear for the global audience, then why is there so much disagreement among the global audience?

Page 1, Line 26: "For the coming decades..."; I believe this should be "In the coming decades..."

Page 1, Line 28: "The reduction in ice volume may produce an array of consequences which requires to be properly predicted." This sentence, and the following, is vague. What consequences are you talking about? Be explicit.

Page 2, Line 2: "For any glacier model..."; Saying "any" makes this sentence too broad and not necessarily true. Be explicit for the classes/types/purposes of modes which require SMB and glacier dynamics (e.g. "SMB and glacier dynamics both need to be modelled to understand glacier evolution on regional and sub-regional scales. Models of varying complexity exist for both processes.")

Page 2, Line 18: "... these different approaches strongly depend on available data..."; Change to "... the use of these different approaches strongly depend on available
data..." since it the model usage, not the model itself, which depends on what data one has.

Page 2, Line 21: "... relationships remain stationary."; Change to "... relationships remain stationary in time."

Page 2, Line 34: "... the glaciological community has remained quite oblivious to these advances..."; Oblivious is a strongly negative word to use here, and it is a disservice to insult your readers.

Page 6, Line 10: "... relevant predictors must be selected, performing a sensitivity study..."; Change to "... relevant predictors must be selected, so we perform a sensitivity study..."

Page 6, Line 14 and Equation 1: Is there a reference for this "effective way of expanding the training dataset"?

Page 7, last sentence: Here you describe the types of cross validations available in ALPGM. Which did you use?

Page 8, Line 6: "... (2) the optimizer: the method for..."; change to "...(2) the optimizer, which is the method for..."

Page 8, Line 6: "...(3) its (possibly nonlinear) activation functions..."; When are activation functions linear?

Page 8, Line 10: "... allowing to train deep neural networks..."; change to "allowing the training of deep neural networks..."

Page 8, Line 11: "...ANNs are best suited when the quality of predictions prevails over the interpretability of the model." This is vague, and does not help readers know when ANNs are 'best suited'. How are either of these things quantified?

Page 10, Line 21: "...should be long enough to be representative of the glacier evolution..."; How long is 'representative'? Representative of what? How does one
know this?

Page 11, Line 3: Refer to Figure 4 here.

Page 11, Line 5: Is there a reference to this study?

Page 11, Line 9: "... using remote sensing based on changes in glacier volume and the snow line altitude is used..."; Remove second "is used"

Page 12, Figure 4: Axes should be labelled.

Page 13, Line 2: Cite RGI (check here for reference: https://www.glims.org/RGI/)

Page 13, Line 12: Qualifications here are vague (e.g. "quite satisfactorily", "good overall", "certain altitudinal ranges"). Give quantitative measures of "goodness", and refer to specific parts of Figure S2 that demonstrate what you're talking about.

Page 13, Line 26: This sentence can be improved by maintaining consistency across clause structure. You use "we verb" statements (e.g. we go through, we assess, and we show) for all clauses except for "the building of the machine learning SMB models".

Page 14, Line 25 (and paragraph): You discuss that you dynamically calculate the accumulation/ablation periods based on the CPDD, and that you keep constant periods to account for winter and summer snowfalls. Later, you use 'transition months' as predictors – are these predictors kept constant, or dynamically calculated? Are results improved when the transition months are dynamically computed? I ask because I would expect that what constitutes a 'transition month' may change in the future. Or do you think that this approach, applied to more variables, then forces the model to depend too much on CPDD when the CPDD is not the only variable involved in melt?

Page 15, Equations 2 and 3: Are input variables normalized? If so, how?

Page 15, Line 20: When you say 'linear machine learning', are you referring to the linear regression methods? Be consistent in how you refer to your methods.
Page 15, Line 20: How did you choose the function f?

Page 15, Equation 25: You create linear models using the predictors shown here. You then create nonlinear models using only the predictors in Equations 2 and 3. Then, you compare the results of these models and conclude that the nonlinear model is better because of the nonlinear nature of the model; however, how do you know that the improved performance is not simply due to using a different set of predictor variables? Your argument would be more convincing if you first showed that the linear model performance improved when you change predictor variables from the standard case (those only in Equations 2 and 3) to the combination case (Equation 4), and then showed that a nonlinear model using variables from the standard case outperformed even this improved linear model.

Page 15, Equation 5: Is there a missing '('? This equation ends with ')\_g,y'.

Page 16, Line 1: It is not clear to me why there are 50 predictors, when there are 33 coefficients in Equation 4.

Page 16, Line 2: Can you please be more explicit about what this matrix is, and to what matrix equation it is input into?

Page 16, Line 8: "...the annual CPDD as well as the winter and summer snowfall appear as significant predictors as well as several monthly mean temperatures and snowfall values..."; Change to "...the annual CPDD, winter and summer snowfall, and several monthly mean temperature and snowfall were found to be significant at p

Page 20, Line 15: "This implies, that..."; remove the comma

Page 22, Line 7: "...using Leave-Some-Glaciers-and-Years-Out (LSYGO)"; the abbreviation should be LSGYO, or the full phrase should be Leave-Some-Years-and-Glaciers-Out, for consistency.

Page 25, Lines 4-5: "The greater the dropout, the more we will constrain the learning of the ANN so the higher the generalization will be, until a certain point." This sentence is not clear. What does it mean to "constrain the learning"? Why is there a "certain point", and what happens beyond that point? This could be made more explicit.

Page 25, Line 7: Why is it that slower convergence leads to better generalization? Is this always true?

Page 25, Line 8: "... that best suits a certain dataset and model." How does one define "best"?

Page 25, Line 17: "Despite it has been shown..."; Change to "Although it has been shown" or "Despite the fact that it has been shown"

Page 25, Line 28: "The results were quite astonishing..."; If the results are astonishing, then this result warrants further emphasis in the paper. The methods used to come to this conclusion should be brought up in Section 3, and further discussion is warranted in Section 4. It is worth a figure to communicate these results.

Page 26, Lines 5-16: This paragraph is speculative, but is presented with a high degree of confidence. Phrases such as "unprecedented efficiency" and "excellent" are used without supporting evidence. Much of the discussion is implied; for example: "An interesting way of expanding a dataset would be to use a deep learning approach to fill the data gaps." It is unclear how this would be done. "Such an approach would be an excellent way of obtaining more SMB data in remote glacierized regions such as the Andes or the Himalayas." This is not known or demonstrated by the rest of the paper. I would recommend either removing this paragraph entirely or severely limiting

TCD
its scope.

Page 26, Lines 18-22: This paragraph is speculative. It does not follow from the results presented in the paper, and is more of a justification for using deep learning in glaciology than it is an item of discussion in the context of the preceding research. These final two paragraphs do a disservice to the rest of the paper; prior to this, the organization had nice flow, and the first two paragraphs in Section 4.3 were both interesting and directly relevant.

Page 27, Lines 4-12: This paragraph is quite vague and does not explicitly follow from the research. For example: "It might still be too early for the development of such models in certain regions" is a vague statement. Conclusions should follow directly and explicitly from the work and should not reach beyond the scope of the research. The first paragraph in Section 5 is much better.

Supplementary Figures: I the SMB\_lasso\_ANN\_no\_weights\_SMB\_simulations.pdf file, y-axes are missing units.

Figures 6, 7, 9, and 11: Please increase font size, especially of axis labels.

---

## Referee Comment (RC2) · Fabien Maussion (Referee) · 8 Sep 2019

**Review of: "Deep learning applied to glacier evolution modelling" by Bolibar and co-authors**

**1   General comments**

In this study, Bolibar and co-authors train various machine learning algorithms to compute the surface mass balance (SMB) of 32 glaciers in the French Alps. The paper is well written, timely (machine learning is a trendy topic and will continue to be so in the future), and is an interesting read. Its focus is solely on model development and performance, without application or discussion of model output: therefore, "Geoscientific Model Development" could have been a better venue for such a study.

I am sympathetic towards the main premise of the paper, which is to demonstrate the suitability of deep-learning for SMB modelling, and the open-source tools provided with the paper further increase the relevance of the paper as an example for future studies to build upon. However, I have some concerns with the current (unclear) focus of the study and certain methodological aspects, which I believe need to be addressed before publication.

**1.1   GC1: the use of glaciological predictors**

Currently, the statistical models have the possibility to train on certain topographical predictors (more specifically: Mean glacier altitude, Slope of the lowermost 20% glacier altitudinal range, Glacier surface area). These predictors are time-dependant and extracted from DEMs and inventories at various times during the study period. Regardless of the fact that these data are very unlikely to be available in such a precision for many other regions of the world (and certainly not for past and future glacier states outside the observation period), using them as explanatory variables poses a serious conceptual problem: these variables are meant to be **simulated** by the full model (SMB + glacier evolution), thus contradicting the need for a glacier evolution model in the first place. I see three ways out of this chicken and egg problem, all with drawbacks and likely to affect the accuracy of the model:

- use time-independent predictors such as a constant area (probably a bad idea because this will raise model validity problems for longer simulations)

- show that your full model is able to simulate those, and then use the modelled ones as input data for the next year in the "model application period" (i.e. your statistical model will have to be called in yearly time steps). This is possible but will require some thinking about how to validate the procedure.

- don't use them at all (simplest)

Regardless of your choice, the study will have to be adapted to this change. Note that I saw that the predictors weren't chosen by the Lasso model, but: (i) you don't know if they aren't chosen by the cross-validation models, and (ii) because I'm unsure how the predictor selection for the ANN really works I don't know if they play a role there.

**1.2  GC2: glacier wide mass-balance**

The model is trained to reproduce glacier wide mass-balance (or "specific MB"). Glacier wide mass balances are dependent on the altitude-area distribution of the glacier and therefore are not only dependant on climate but also on the glacier's dynamical response to current and past climates. This has been discussed elsewhere and in another context (e.g. `https://doi.org/10.5194/tcd-4-2475-2010`) and there are good arguments for both sides, but I still can't believe that predicting glacier wide mass-balance is a good idea for a glacier evolution model. For example, consider this idealized glacier response to a step climate change:

[Figure]

Source and context: `https://oggm.org/2017/10/01/specmb-ela`

In this perfectly linear SMB framework (linear gradient, linear response to step change), the simplest of the statistical models could simulate the fixed-geometry SMB (or even any point SMB) *perfectly*, but it would completely fail to simulate glacier-wide SMB, which requires knowledge about past glacier states and evolution. This is an extreme case, but still raises questions about this study (even in the relatively short period considered here).

The large-scale glacier evolution models I am aware of use either an altitude-dependant SMB (e.g. OGGM, GloGEM, PyGEM) or parametrize this non-linear response in their SMB model (Marzeion et al., 2012). I think that it is too late to change this in your framework at this time, but I strongly recommend to explore other approaches for future studies based on ALPGM. If your model ever intends to simulate many glaciers over long periods, I think that this effect should be treated explicitly (or it should be shown that the "black-box" ANN can properly deal

with the full-glacier problem as suggested in the discussion section). Regardless of your choice, this point needs to be discussed in the paper.

**1.3   GC3: focus of the study**

In my understanding, this study attempts to make three points:

1. Machine learning (and deep-learning in particular) is a useful tool for glaciology

2. Introduce and validate a new SMB model based on deep-learning

3. Introduce a new glacier evolution model (ALPGM)

While I think that the study is fairly successful for points 1 and 2, it does not succeed for point 3. My concerns about point 3 are strongly driven by methodological considerations (GC1 and GC2 above, the use of a perfectly fitted $\Delta h$ method impossible to validate, and the lack of proper out-of-sample validation of the full ALPGM model). This confusion about the goals of the study also make the paper's introduction and title quite confusing. I would much rather see this study focus on point 1 and 2 (for which you provide tangible results and arguments) and remove point 3 (and the corresponding section "3.3 Glacier geometry evolution: validation" which, in the authors own words, isn't the main focus of the study). Removing point 3 would help to focus on the strength of the current version of ALPGM as a mass-balance model. If the author's choose to keep point 3, then I have several concerns about whether ALPGM really is a glacier evolution model (yet).

**2   Specific comments**

**Abstract L22** : "for past and future climates.": remove "future", since this has not yet be demonstrated.

**P2 L6-16** : although it is tempting to classify the models like this, I think that this list (and several other parts of the introduction) needs more precise definitions and a clearer positioning of the ALPGM model. In this list, you need to differentiate between the treatment of ice flow / glacier evolution by these models (on which your classification seems to be based, but not explicitly so) from the treatment of surface mass-balance (SMB), which is what your study is actually about. The "Physics-based models" that you list in fact often have no SMB module, and rely on external SMB as an external boundary condition in real-world applications. For the sake of clarity and given the scope of your study, I would rather focus on the hierarchy of SMB models (with SEB or even coupled Atmo-SEB models being the more advanced, and temperature index models the simpler models). Please rethink this part of the introduction, as well as the following paragraph.

**P2 L34** : "Compared to other fields in geosciences": which ones?

**P2 L34** : "the glaciological community has remained quite oblivious to these advances": this is a subjective statement, you need to mention that this is your opinion.

**P3 L8** : "but all of them were linear, which are not necessarily the most suitable for modelling the nonlinear climate system": you have a very "statistical" view of linearity here. The

statistical models used in Maussion et al. are linear, yes, but they target individual SEB fluxes which are then transformed to be physical (e.g. by preventing negative precipitation or a non-closed SEB budget) and then used to compute the SMB. As a result, the full model $M$ (as in $SMB = M(y)$ with $y$ the predictors and $SMB$ the target variable) is non-linear. This is important also in the context of traditional temperature index or degree day models, which can be compared to linear models applied to transformed predictors and as such, are also non-linear (e.g. by preventing melt for negative temperature or by transforming precipitation to solid precipitation). This is an important feature: without this non-linearity, they wouldn't work at all.

**P4 L15** : "When most glacier models tend to incorporate more and more physical processes (Maussion et al., 2019; Zekollari et al., 2019), ALPGM takes an alternative approach based on data science." Are you talking about SMB or ice dynamics? Your "data-science" is applied to the SMB problem here, and I believe it would be more appropriate to cite models of SEB/SMB in this sentence (e.g. Hock et al, Mölg et al, CROCUS, or similar).

**P5 L11** : "leave-one-glacier-out (LOGO) or a leave-one-year-out (LOYO) cross-validation". Congratulations for coining these - I wish we had invented these acronyms earlier.

**P5 L29-34** "Although the features used as input (...) will likely have different biases." This paragraph seems out of context here and should be moved to the discussion

**P6 L25** : "StatsModel" is spelled "StatsModels"

**P8 L1** : "The generated coefficients from the model serve to determine the significant predictors to be kept for the artificial neural network training." is Lasso part of the feature selection process of the ANN then? This raises interesting (and hard) questions concerning cross-validation and the model's real independence from training data. Furthermore, it gives an advantage to ANN over the linear models since their predictors are pre-filtered (see e.g. the "double Lasso" method which makes this an advantage as well). Please comment.

**P9 L5-17** : hyperparameters. As a non specialist of "deep-learning", I need to ask: shouldn't this hyperparameter selection also be cross-validated? In Lasso, for example, the regularization parameter could be called an "hyper-parameter" and its selection takes place within the model tuning step, effectively making any external cross-validation (realized by LOGO and LOYO in your case) a "true" out-of-sample validation. What about the ANN hyperparameters? Please comment.

**Glacier geometry update** You call the geometry update a "parametrization" but in my opinion it isn't: you use an empirical $\Delta h$ function perfectly known for each glacier since it is individually fitted. A true "parameterization" (like the one used in Huss and Hock 2015) would have the goal to work for any unseen glacier. Currently your model cannot be applied )(or validated) against unseen glaciers.

**Figure 4** : Since you have DEMs (and geodetic MBs) from all blue glaciers in Figure 4, can you apply your model to them as well and compare? This would be a good (but partial) out-of-sample validation ("partial" because you still need knowledge about the glacier's $\Delta h$).

**Glacier ice thickness** : To avoid confusion: if still applicable after revision, mention here that ice-thickness are only used for the 2003-2016 test run, and not for the rest of the model workflow.

**Glacier topographical variables** : from an email question to the authors I know that the topographical predictors (e.g. area, slope) are time-dependant and obtained from various DEM snapshots. This needs to be explained here. Regardless of this missing explanation, this raises questions about the overall applicability of the method to unseen situations (see general comment).

**P16 L2** : "For the training of the ANN, no combination of topo-climatic features is done as previously mentioned". I have a hard time finding where this is explained. Is this the part with Lasso? In any case, the predictor selection for ANN needs to be explained here for consistency and to help the reader.

**P16 L9** : "Latitude and longitude seem to play an important role when combined with snowfall.". I don't really understand the climatological explanation that follows this statement. If the reanalysis data is accurate, then these east-west and north-south differences should already be in the training data. If anything, these combination of predictors play the role of bias correction - or are the result of luck (which is often the case with many co-linear predictors).

**Lon/Lat Predictors** : this is a subjective opinion, but I suggest to remove Lon/Lat predictors from the set. They should not explain anything which isn't in the climate and topographical predictors already, and using lon/lat seriously hinders the applicability of the model to larger areas.

**Figs 6 and 7** : although visually appealing, the use of different colorscales for the ANN and linear models is misleading. All four plots are exact same and should have the same colorscale, min-max range, x-y axis, etc.

**Figs 6 and 7 and corresponding discussion about explained variance** : a possible improvement to describe the models errors is to plot binned model error (residuals) as a function of the target variable (here, SMB), or use a Q-Q Plot. It would display in a more quantitative way the non-normal distribution of model errors (visible on the scatter plots by a flattening on both ends of the scatter), further making your point that ANN is better (but not perfect) at reproducing the true variance of the data.

**Figure 10** : a striking feature of figure 10 and not discussed in the manuscript is the clear tendency of LASSO to overestimate MB in the second half of the period and underestimate MB in the first half. I guess it is a result of more frequent negative MBs in the second half, which are underestimated by the model with an obvious lower variance, but is this the only reason?

**Glacier geometry evolution validation** : these results are not too surprising. Since your evolution model knows exactly where mass is going to be removed (based on data going up to 2011). This test is basically a bias test: if the model has no bias, ice is going to be removed at the right place (because you know where to remove it) and the area will be correct, provided that the ice thicknesses are more or less accurate.

**P24 L1** : "Even for a 12-year period, the initial ice thickness remains the largest uncertainty": this statement is not supported by your results, since this is the only uncertainty you consider in Fig. 11. Here, you could add model uncertainty by using out-of-sample validation (by training the model with data only before 2003 and using LOGO), or use uncertainty measures derived by cross-validation. The issue with the $\Delta h$ method raised above would remain, though.

**P25 L26** : "we trained an ANN only with monthly average temperature and snowfall, without any topographical predictors". These experiments should become the central component of your study, not the other way around (see general comment).

---

## Author Response (AR1)

**Fabien Maussion**

**1 General comments**

In this study, Bolibar and co-authors train various machine learning algorithms to compute the surface mass balance (SMB) of 32 glaciers in the French Alps. The paper is well written, timely (machine learning is a trendy topic and will continue to be so in the future), and is an interesting read. Its focus is solely on model development and performance, without application or discussion of model output: therefore, Geoscientic Model Development" could have been a better venue for such a study.

I am sympathetic towards the main premise of the paper, which is to demonstrate the suitability of deep-learning for SMB modelling, and the open-source tools provided with the paper further increase the relevance of the paper as an example for future studies to build upon. However, I have some concerns with the current (unclear) focus of the study and certain methodological aspects, which I believe need to be addressed before publication.

We would like to thank Fabien Maussion for his thorough, constructive and generally positive review, which raised very interesting questions regarding glacier modelling. We believe that it has helped to revise certain methodological aspects as well as to bring interesting new elements in the discussion. All comments from the review have been responded individually.

Regarding the journal choice, as it will be later discussed in GC3, "The Cryosphere" has been chosen over "Geoscientific Model Development" in order to focus more on the application and feasibility of a machine learning approach for SMB reconstruction and simulation, coupled with a glacier geometry update component, rather than the presentation of a new glacier evolution model. ALPGM has been developed as a tool during the first year of the main author's (Jordi Bolibar) PhD thesis. Therefore, we believe that the interest of the chosen approach (deep learning SMB modelling) prevails over the general interest of the model and code itself. Since the machine learning SMB modelling approach requires to re-train the model(s) for other regions, the most interesting part is the methodology itself, and not the fitted parameters that are, after all, specific to each geographical region. ALPGM being open-source, can be easily re-used, but a limited amount of time was dedicated to creating easily re-usable code and interfaces, needed for a readily shareable glacier model. This kind of purely software engineering tasks are extremely time consuming, as the reviewer will know from his own experience with OGGM. Therefore, we preferred to discuss the approach itself (the main topic of the article) which can be re-used by everyone (with or without our code) rather than making the presentation a model which is far from being "plug-and-play" applied to other regions. In that sense, the aim of this paper is to present the approach and methodology that will be used in future studies to simulate SMBs and the evolution of glaciers in the French Alps (the goal of Jordi Bolibar's PhD thesis).

**1.1 GC1: the use of glaciological predictors**

Currently, the statistical models have the possibility to train on certain topographical predictors (more specifically: Mean glacier altitude, Slope of the lowermost 20% glacier altitudinal range, Glacier surface area). These predictors are time-dependant and extracted from DEMs and inventories at various times during the study period. Regardless of the fact that these data are very unlikely to be available in such a precision for many other regions of the world (and certainly not for past and future glacier states outside the observation period), using them as explanatory variables poses a serious conceptual problem: these variables are meant to be simulated by the full model (SMB + glacier evolution), thus contradicting the need for a glacier evolution model in the first place. I see three ways out of this chicken and egg problem, all with drawbacks and likely to affect the accuracy of the model:

(a) use time-independent predictors such as a constant area (probably a bad idea because this will raise model validity problems for longer simulations)

(b) show that your full model is able to simulate those, and then use the modelled ones as input data for the next year in the "model application period" (i.e. your statistical model will have to be called in yearly time steps). This is possible but will require some thinking about how to validate the procedure.

(c) don't use them at all (simplest)

Regardless of your choice, the study will have to be adapted to this change. Note that I saw that the predictors weren't chosen by the Lasso model, but: (i) you don't know if they aren't chosen by the cross-validation models, and (ii) because I'm unsure how the predictor selection for the ANN really works I don't know if they play a role there.

These aspects of the functioning of the model and its specific behaviour appear not to be clear enough in the manuscript, so we reworded parts of the manuscript to clarify ALPGM's approach when dealing with the topographical predictors. In a nutshell, ALPGM already uses the technique (b) suggested by the reviewer, but only for the "glacier evolution" component: it learns from interpolated topographical data, as predictors are not necessarily available at each time point, but it is able to simulate glacier-wide SMBs iteratively year-by-year by propagating a newly computed glacier-wide SMB in the model to generate updated topographical predictors for the next year. In more details:

1. In the "glacier-wide SMB modelling" component, which works totally independently from the "glacier evolution" component, the glacier-wide SMB machine learning models are trained based on the historical data available. Topographical and climate data for every year and glacier available in the region of interest is collected. As shown in our case study of French alpine glaciers, the climate data comes from the SAFRAN reanalysis, which is computed for each glacier at its centroid, and the topographical predictors come from the 1967, 1985, 2003 and 2015 glacier inventories (Gardent et al., 2014, with 2015 update). In order to have topographical data for each year, the topographical variables are linearly interpolated between inventories. This is indeed an

approximation and a hypothesis, but as we show throughout the extensive cross-validation, such approximation is enough for the model to understand the relationship between these variables and the glacier-wide SMB. With all this data, the machine learning glacier-wide SMB models are trained and cross-validated using LOGO, LOYO and LSYGO. Therefore, glacier-wide SMB machine learning models are trained with all the training data at once, and then tested on the residual test data for each of the cross-validation folds. After cross-validation, a SMB model is chosen for the spatial, temporal and spatiotemporal dimensions and it is stored as a file which will be later called in the "glacier evolution" component.

2. In the "glacier evolution" component, responsible for coupling the simulated glacier-wide SMBs and the geometry update, these topographical variables are computed differently in order to allow the simulations to be carried out for time periods and glaciers outside the historically observed ones. For each glacier in the region to be simulated, Farinotti et al. (2019) ice thickness and DEM glacier-specific rasters are retrieved for the starting date of the simulation (2003 in our study case). Then, in a loop, for every glacier and year, the topographical predictors are computed from these raster files. Then, the climate predictors at the glacier's current centroid are retrieved from the climate data (e.g. reanalysis or projections) and with all this data the input topo-climatic data for the glacier-wide SMB is assembled. Then, the glacier-wide SMB for this glacier and year is simulated, which combined with the glacier-specific Δh function allows to update the glacier's ice thickness and DEM rasters. This process is repeated in a loop, therefore updating the glacier's geometry with an annual timestep and taking into account the glacier's morphological and topographical changes in the glacier-wide SMB simulations. For the simulation of the following year's SMB, the previously updated ice thickness and DEM rasters are used to re-calculate the topographical parameters, which in turn are used as input topographical predictors for the glacier-wide SMB machine learning model.

This iterative process of re-calculation of topographical predictors before the simulation of the glacier-wide SMB for every year is done in the glacier evolution component, which is intended to be used for glaciers and time periods from the year 2003 onwards. Indeed, since the ice thickness database is based on this year, full simulations with the coupling of the glacier-wide SMB component and the glacier evolution component can only be run from this year onwards. This constraint is however not important for the goal of our general project, which is to be able to simulate the future evolution of French alpine glaciers.

Indeed, few regions in the world have the wealth of data available in the French Alps, especially regarding the multitemporal glacier inventories. Nonetheless, these could be obtained differently in other regions, using global or multiple DEMs covering the region of interest. Moreover, as it is discussed in GC2, the glacier-wide SMB model has proved to still work without any topographical predictors. This aspect is discussed in detail in our reply to GC2.

Regarding Fabien Maussion's comment on the predictor selection procedure, it is discussed in detail in the Specific comments from P8 L1 and P16 L2.

In order to increase the clarity on how the topographical predictors are computed and their different sources, the following changes have been made:

- In Sect. 2.1 "Model overview and workflow":

"2. The SMB machine learning component retrieves the preprocessed **climate predictors from the stored files, retrieves the topographical predictors from the multitemporal glacier inventories**, and then it assembles the training dataset by combining all the necessary topo-climatic predictors. A machine learning algorithm is chosen for the SMB model, which can be loaded from a previous run or it can be trained again with a new dataset. **Then, the SMB model(s) are trained with the full topo-climatic dataset**. These model(s) are stored in intermediate files, allowing to skip this step for future runs."

[..]

"5. Once all previous steps have been run, the **glacier evolution** simulations are launched. For each glacier, the initial ice thickness **and DEM rasters and the glacier geometry update** function are retrieved. **Then, in a loop, for every glacier and year, the topographical data is computed from these raster files. The climate predictors at the glacier's current centroid are retrieved from the climate data (e.g. reanalysis or projections) and with all this data the input topo-climatic data for the glacier-wide SMB model is assembled. Afterwards, the glacier-wide SMB for this glacier and year is simulated, which combined with the glacier-specific geometry update function allows to update the glacier's ice thickness and DEM rasters. This process is repeated in a loop, therefore updating the glacier's geometry with an annual timestep and taking into account the glacier's morphological and topographical changes in the glacier-wide SMB simulations. For the simulation of the following year's SMB, the previously updated ice thickness and DEM rasters is used to re-calculate the topographical parameters, which in turn are used as input topographical predictors for the glacier-wide SMB machine learning model.** If all the ice thickness raster pixels of a glacier become zero, the glacier is considered as disappeared and is removed from the simulation pipeline. For each year, multiple results are stored in data files as well as the raster DEM and ice thickness values for each glacier."

- Fig.1 in Sect. 2.1 "Model overview and workflow", has been updated to indicate the fact that the annually updated topographical predictors are used as input for the glacier-wide SMB model:

[Figure]

- In Sect. 3.1.2 "Topographical glacier data and altimetry", the following changes have been added to clarify the source of topographical data at different steps of the model:

"**The topographical data used for the training of the glacier-wide SMB machine learning models is taken from the multitemporal inventory of the French Alps glaciers** (e.g., Gardent et al., 2014) partly available through the GLIMS Glacier Database (NSIDC, 2005). We worked with the 1967, 1985, 2003 and 2015 inventories (Gardent et al., 2014, with 2015 update). **Between these dates, the topographical predictors are linearly interpolated. On the other hand, in the glacier evolution component of ALPGM (Fig.1, step 5), this topographical data is re-computed every year for each glacier from the evolving and annually updated glacier-specific ice thickness and DEM rasters (Sect. 3.1.3). Since these raster files are estimates for the year 2003 (Farinotti et al., 2019 for the ice thickness), the full glacier evolution simulations can start the earliest at this date. For the computation of the glacier-specific geometry update functions, two** DEMs covering the whole French Alps have been used: (1) one from 2011 generated from SPOT5 stereo-pair images, acquired on 15 October 2011; and (2) a 1979 aerial photogrammetric DEM from the French National Geographic Institute (Institut Géographique National, IGN), processed from aerial photographs taken around 1979. Both DEMs have an accuracy between 1 and 4 meters (Rabatel et al., 2016), and their uncertainties are negligible compared to many other parameters in this study."

- Finally, in order to prove that the model, combining the glacier-wide SMB model and glacier evolution components, can successfully simulate the evolution of the topographical parameters and their feedback we have performed a specific

test for this. From the year 2003 until 2014, we have run the glacier-wide SMB simulations for the 32 case study glaciers, first with the topographical predictors coming from the interpolated multitemporal glacier inventories (used during the training of the SMB model), and then with the full glacier evolution model, with the Farinotti et al. (2019) ice thickness and DEM raster files. When comparing the results of these simulations, their differences are minimal, mostly coming from the differences between input data (higher resolution DEMs and data from the glacier inventories vs. lower resolution data from Farinotti et al. 2019). For the 2003-2014 period, we obtained very similar performances: an average RMSE of 0.49 m.w.e. a$^{-1}$ using the multitemporal glacier inventories and an average RMSE of 0.52 m.w.e. a$^{-1}$ using the full glacier evolution model with the Farinotti et al. (2019) raster data.

This test, its results and the discussion elements have been added as a new section in the supplementary material, plus a new figure (S6):

**3. Topographical glacier-wide SMB predictors**

Since topography plays a role in the glacier-wide SMB signal, besides the climate, the representation of the glacier's topography is important in order to correctly simulate its glacier-wide SMB and its geometrical evolution. As explained in Sect. 2.1 "Model overview and workflow" and Sect. 3.1.2 "Topographical glacier data and altimetry", the source of the topographical predictors used for the simulation of glacier-wide SMB is different at different steps of the glacier evolution simulation chain. Two cases exist:

1. For the machine learning training of the glacier-wide SMB models, which is performed on historical data, all topographical data comes from the multitemporal glacier inventories (Gardent et al., 2014, with 2015 update). In order to have an annual timestep, topographical data from these inventories are linearly interpolated.

2. For the full glacier evolution simulation, coupling the glacier-wide SMB component with the glacier geometry evolution component, the model must be capable of generating all the input topographical predictors even for non-observed glaciers and future periods. For every glacier and year, all the topographical predictors are computed from the updated glacier-specific ice thickness and DEM raster files from Farinotti et al. (2019), which then are used to simulate a single glacier-wide SMB for that glacier and year. Then, this glacier-wide SMB together with the glacier-specific geometry update function are used to update the glacier's geometry and their respective ice thickness and DEM rasters. For the next year, all the topographical predictors are recomputed with the updated raster files, and this process is repeated in a loop with an annual timestep. Therefore, the glacier-wide SMB model is called with an annual timestep, simulating only single values in order to take into account the evolution of the glacier's topography.

In order to show that the glacier geometry update component, coupled with the glacier-wide SMB simulation component can successfully simulate the evolution of the topographical characteristics of glaciers in the region, a specific test was designed. Using the same validation period as in Sect. 3.2 (2003-2015), we ran parallel simulations of

glacier-wide SMB for all the 32 case study glaciers. The first simulation was done using case (1) with the multitemporal glacier inventories data, and the second one was done following case (2) with the full glacier evolution model and the Farinotti et al. (2019) raster files. The results of both simulations were really similar, revealing only small differences. On average, the simulated glacier-wide SMBs for this period differed on 0.069 m w.e. a$^{-1}$, due to the differences in the input topographical predictors, which are computed from different datasets (Fig. S6). Moreover, the performances of both simulations for this period are very similar, with a RMSE of 0.49 m.w.e. a$^{-1}$ for case (1) and 0.52 m.w.e. a$^{-1}$ for case (2). The results with all the differences between the simulated glacier-wide SMB values and input topographical values are summarized in Table S1:

| Variable (multitemporal inventories vs. full glacier evolution) | SMB simulated | Slope | Average glacier elevation | Area |
|---|---|---|---|---|
| **Mean difference** | 0.069 m.w.e a$^{-1}$ | 1.8° | 31.3 m | 0.2 km$^2$ |

**Table S1:** Differences on simulated glacier-wide SMB and topographical predictors between a simulation using interpolated topographical predictors from the multitemporal glacier inventories and the full glacier evolution simulations including the coupling of the glacier-wide SMB with the glacier geometry update.

The only striking difference is perhaps the difference in simulated areas. This is mainly due to the fact that the Farinotti et al. (2019) dataset uses the RGI v6, which for the largest glaciers of Argentière and Mer de Glace, overestimates its surface area (from 32 to 34 km$^2$ for Mer de Glace in 2003). The differences in slope are explained by the fact that this variable is not included in the multitemporal glacier inventories (Gardent et al., 2014), therefore it has been computed once with a global DEM and kept constant for each glacier throughout the years for the training of the SMB model. On the other hand, in order to include the long term effects of glacier morphology changes in the glacier evolution simulations (glacier-wide SMB simulation + glacier geometry update), the glacier slope is re-computed with an annual timestep and it evolves through time. Therefore, there are small differences for certain glaciers whose slope has evolved during this period, thus accounting for the differences with the fixed value used for the training of the SMB model.

This test serves to prove that the full glacier evolution simulations in ALPGM are capable of reproducing the topographical predictors used for the training of the glacier-wide SMB machine learning models. Moreover, this test also helps to prove that ALPGM can correctly simulate the topographical evolution of glaciers, which allows to capture the topography induced feedback, which plays a role in the simulation of glacier-wide SMBs.

[Figure]

**Figure S6:** Comparison of glacier-wide SMB simulations (2003-2015, 32 case study glaciers) using topographical predictors from the multitemporal glacier inventories (Y axis) vs. using the full glacier evolution simulations in ALPGM with the Farinotti et al. (2019) ice thickness and DEM rasters (X axis). Average difference = 0.069 m.w.e. a$^{-1}$

**1.2 GC2: glacier wide mass-balance**

The model is trained to reproduce glacier wide mass-balance (or "specific MB"). Glacier wide mass balances are dependent on the altitude-area distribution of the glacier and therefore are not only dependant on climate but also on the glacier's dynamical response to current and past climates. This has been discussed elsewhere and in another context (e.g. https://doi.org/10.5194/tcd-4-2475-2010) and there are good arguments for both sides, but I still can't believe that predicting glacier wide mass-balance is a good idea for a glacier evolution model.

For example, consider this idealized glacier response to a step climate change:

Source and context: https://oggm.org/2017/10/01/specmb-ela

In this perfectly linear SMB framework (linear gradient, linear response to step change), the simplest of the statistical models could simulate the xed-geometry SMB (or even any point SMB) perfectly, but it would completely fail to simulate glacier-wide SMB, which requires knowledge about past glacier states and evolution. This is an extreme case, but still raises questions about this study (even in the relatively short period considered here).

The large-scale glacier evolution models I am aware of use either an altitude-dependant SMB (e.g. OGGM, GloGEM, PyGEM) or parametrize this non-linear response in their SMB model (Marzeion et al., 2012). I think that it is too late to change this in your framework at this time, but I strongly recommend to explore other approaches for future studies based on ALPGM. If your model ever intends to simulate many glaciers over long periods, I think that this effect should be treated explicitly (or

it should be shown that the "black-box" ANN can properly deal with the full-glacier problem as suggested in the discussion section). Regardless of your choice, this point needs to be discussed in the paper.

This comment raises interesting questions on many levels. As the reviewer mentions, the discussion of glacier-wide *vs.* point/altitude-dependent SMB is a widely discussed topic in the glaciology community. Indeed, glacier-wide SMBs contain not only information on climate but also on the glacier's surface area and topography. Therefore, if a statistical model attempts to simulate glacier-wide SMBs based solely on climate data, it will be missing some information. How important this information actually is, is the main topic of debate, addressed in the two comments from Leclercq et al. and Huss et al. on Huss et al. (2010), as well as in Huss et al. (2012) "Conventional versus reference-surface mass balance".

In most glacier evolution models, SMBs are simulated at different altitudes in order to integrate them in the spatialized modelling framework which takes into account glacier dynamics and ice flow. Nonetheless, this is not necessary when using a glacier geometry update parameterization like the Δh parameterization, as only the total glacier mass annual variation is needed to be redistributed by the Δh function. Therefore, here we chose to work with glacier-wide SMB data due to the fact that in the French Alps there is much more glacier-wide SMB data available (Rabatel et al. 2016) than altitude-dependent or point SMB, essential for training machine learning models. In order to correctly simulate glacier-wide SMB, we included topographical variables that intervene in determining the glacier-wide SMB of glaciers in the French Alps, based on a literature review and on a sensitivity statistical analysis (Sect. "3.2.1 Selection of predictors"). Consequently, our glacier-wide SMB machine learning models do take into account these topographical parameters.

In order to verify these concerns raised by the reviewer, we remade from scratch the causal analysis using Lasso, but this time, no combination of topo-climatic predictors was used. We believe that using linear combinations of topo-climatic predictors increases a lot the number of input predictors, which in turn reduces their relative weight, making the causal analysis and the statistical inference more difficult. Moreover, since the ANN does not take topo-climatic combinations of predictors as input, this new subset of predictors used for the causal analysis is more representative, despite being linear, of the relationships the ANN must be using internally. In this new analysis, the results are similar to the previous causal analysis (Fig. 5). Climate predictors still appear, by far, as the most relevant predictors, with accumulation-related predictors having more importance than ablation-related predictors. Topographical predictors appear to have a minor importance, but they do play a role. Combined, they account for around 3.5% of importance, including the slope of the lowermost 20% altitudinal range, latitude, longitude, area, mean glacier altitude and aspect (Fig. 5). These results are coherent with the spatiotemporal analysis of the glacier-wide SMB signal in the European Alps from Vincent et al. (2017) and Huss (2012), as well as the results from our block cross-validation. The temporal variability of the glacier-wide SMB signal is determined by climate, being the most complex dimension, and topography (differences among glaciers) is modulated by topographical predictors. Therefore, topographical predictors act as spatial

modulators, whereas climate acts as the main signal driver, determining the interannual variability. On top of that, as discussed in Sect. "4.3 Perspectives on future applications of deep learning in glaciology", a LOGO cross-validation was done using no topographical predictors, and it was seen that it had a negative impact on the results, with both the accuracy (RMSE) and explained variance ($r^2$) being negatively affected. This performance penalty (from an RMSE of 0.51 to 0.59 m.w.e. a$^{-1}$), is indeed small, but it is coherent with the importance attributed to topographical predictors in the Lasso causal analysis (Fig.5).

[Figure]

Figure 5. Contribution to the total variance of the 30 top topo-climatic predictors out of 55 predictors using Lasso. Green bars indicate predictors including topographical features, blue ones including accumulation-related features, and red ones including ablation-related features

Therefore, we believe that:

(1) In order to correctly simulate glacier-wide SMBs, topographical predictors play a relevant, albeit secondary, role. They help modulate the glacier-wide SMB signal, introducing spatial changes to the signal determined by climate predictors.

(2) Despite not being possible to "observe" the chosen internal weights and combinations in the ANN, both the Lasso causal analysis and an empirical test removing the input topographical predictors show some benefits in using them. Therefore, it is plausible to conclude that they do play a role in the simulation of glacier-wide SMBs in our case study with French alpine glaciers.

In order to include all these elements of discussion in the manuscript, the following changes have been added:

- **Figure 5** has been updated with the new Lasso causal analysis results.
- The whole Sect. "3.2.2 Causal analysis" has been rewritten to include all these aspects:

"By running the Lasso algorithm on the dataset based on **Eq. 2 and 3**, we obtain the contribution of each predictor in order to explain the annual glacier-wide SMB variance. **Regarding the climatic variables, accumulation-related predictors (winter snowfall, summer snowfall as well as several winter, spring and even summer months) appear as the most important predictors. Ablation-related predictors also seem to be relevant, mainly with CPDD and summer and months at the transition between the seasons (Fig. 5).** Interestingly, meteorological conditions in the transition months are crucial for the annual glacier-wide SMB in the French Alps: (1) October temperature is determinant for the transition between the ablation and the accumulation season, favouring a lengthening of melting when temperature remains positive, or conversely allowing snowfalls that protect the ice and contribute to the accumulation when temperatures are negative; (2) March snowfall has a similar effect: positive anomalies contribute to the total accumulation at the glacier surface, and a thicker snow pack will delay the snow/ice transition during the ablation season leading to a less negative ablation rate (e.g. Fig. 6b, Réveillet et al., 2018). Therefore, meteorological conditions of these transition months seem to strongly impact the annual glacier-wide SMB variability, since their variability oscillates between positive and negative values, unlike the months in the heart of summer or winter.

**On the other hand, topographical predictors do play a role, albeit a secondary one. The slope of the 20% lowermost altitudinal range, the glacier area, the glacier mean altitude and aspect help to modulate the glacier-wide SMB signal which partially depends on glacier topography (Huss et al., 2012) unlike point or altitude-dependent SMB. Moreover, latitude and longitude are among the most relevant topographical predictors, which for this case study are likely to be used as bias correctors of precipitation of the SAFRAN climate reanalysis. SAFRAN is suspected of having a precipitation bias, with higher uncertainties for high altitude precipitation (Vionnet et al., 2016). Since the French Alps present an altitudinal gradient, with higher altitudes towards the eastern and the northern massifs, we found that the coefficients linked to latitude and longitude enhanced glacier-wide SMBs with a northeastern gradient."**

- The discussion elements in Sect. "4.3 Perspectives on future applications of deep learning in glaciology" have been updated:

"**For ALPGM, the SMB machine learning models were trained using glacier-wide SMB data, due to the high availability of glacier-wide SMB data in the French Alps (Rabatel**

**et al., 2016). Nevertheless, the same approach could be used for point SMB data from field observations.**"

[...]

"In this work, we also evaluated the resilience of the deep learning approach: since many glacierized regions in the world do not have the same amount of data used in this study, we trained an ANN only with monthly average temperature and snowfall, without any topographical predictors, to see until which point the algorithm is capable of learning from minimal data. The results were quite interesting, with a coefficient of determination of 0.68 (against 0.76 from the full model) and a RMSE of 0.59 (against 0.51 from the full model). **These results indicate that meteorological data is the primary source of information, determining the interannual variability of the glacier-wide SMB signal. On the other hand, the "bonus" of topographical data helps to modulate the climate signal, by adding a spatial component to better differentiate glaciers and the topographical characteristics included in the glacier-wide SMB data (Huss et al., 2012).**"

1.3 GC3: focus of the study

In my understanding, this study attempts to make three points:

1. Machine learning (and deep-learning in particular) is a useful tool for glaciology

2. Introduce and validate a new SMB model based on deep-learning

3. Introduce a new glacier evolution model (ALPGM)

While I think that the study is fairly successful for points 1 and 2, it does not succeed for point 3. My concerns about point 3 are strongly driven by methodological considerations (GC1 and GC2 above, the use of a perfectly fitted h method impossible to validate, and the lack of proper out-of-sample validation of the full ALPGM model). This confusion about the goals of the study also make the paper's introduction and title quite confusing. I would much rather see this study focus on point 1 and 2 (for which you provide tangible results and arguments) and remove point 3 (and the corresponding section "3.3 Glacier geometry evolution: validation" which, in the authors own words, isn't the main focus of the study). Removing point 3 would help to focus on the strength of the current version of ALPGM as a mass-balance model. If the author's choose to keep point 3, then I have several concerns about whether ALPGM really is a glacier evolution model (yet).

As explained in our reply to "1 General comments", we made the choice to focus the manuscript on presenting the machine learning glacier-wide SMB modelling approach, combined with the parameterized glacier geometry update, rather than presenting a ready-to-use glacier model. Nonetheless, due to some concerns raised in GC1 and GC2, the focus of the model and the manuscript was likely not clear enough.Through the changes proposed in GC1, GC2 and in this section, we hope to improve the clarity, focus and scope of the paper.

We believe that an important source of confusion might come from the fact that ALPGM is actually, in its current design, a regional glacier evolution model that needs to be trained and applied at a regional scale. This information was nonetheless presented in

the manuscript, in the introduction of the machine learning glacier-wide SMB modelling component: since the glacier-wide SMB model needs to be trained with a dataset, the statistical relationships found in datasets are strongest at regional scale, where climate and the glacier's sensitivity to climate remain relatively similar. A more global or continental SMB model could be trained if enough data was available, but nowadays we believe that such an approach is feasible at regional scale only.

- Therefore, ALPGM is now referred to as an **"open-source parameterized regional glacier evolution model"** throughout the manuscript. That is quite a long statement, but at least it is precise.
- In order to set the focus on the approach rather than the presentation of a glacier evolution model, the abstract has been rephrased as follows:

"Abstract. **We present a novel approach to simulate and reconstruct annual glacier-wide surface mass balance series based on a deep artificial neural network (i.e. deep learning). This method has been included as the SMB component of an open-source regional glacier evolution model.** While most glacier models tend to incorporate more and more physical processes, here we take an alternative approach by creating a parameterized model based on data science. Annual glacier-wide SMBs can be simulated **from topo-climatic predictors** using either deep learning or Lasso (regularized multilinear regression), whereas the glacier geometry is updated using a glacier-specific parameterization. We compare and cross-validate our nonlinear deep learning SMB model against other standard linear statistical methods on a dataset of 32 French alpine glaciers. Deep learning is found to outperform linear methods, with improved explained variance (up to +64% in space and +108% in time) and accuracy (up to +47% in space and +58% in time), resulting in an estimated $r^2$ of 0.77 and RMSE of 0.51 m.w.e. Substantial nonlinear structures are captured by deep learning, with around 35% of nonlinear behaviour in the temporal dimension. For the glacier geometry evolution, the main uncertainties come from the ice thickness data used to initialize the model. These results should encourage the use of deep learning in glacier modelling as a powerful nonlinear tool, capable of capturing the nonlinearities of the climate and glacier systems, that can serve to reconstruct or simulate SMB time series for individual glaciers **in a whole region** for past and future climates."

- The same previous changes in the abstract have been reflected in the conclusions:

"**We presented a novel approach to simulate and reconstruct glacier-wide SMB series using deep learning for individual glaciers at regional scale. This method has been included as a SMB component in ALPGM (Bolibar, 2019), a parameterized regional glacier evolution model, following an alternative approach to most physical and process-based glacier models. The data-driven glacier-wide SMB modelling component is coupled with a glacier geometry update component, based on glacier-specific parameterized functions.** Deep learning is shown to outperform linear methods for the simulation of glacier-wide SMB with a case study of French alpine glaciers. By means of cross-validation, we demonstrated how important nonlinear structures (up to 35%) coming from the glacier and climate systems in both the spatial and temporal dimensions are captured by the deep ANN. Taking into account this nonlinearity substantially improved the explained variance and accuracy compared to linear statistical models, especially in the more complex temporal dimension. As we have shown in our case study, deep ANNs are capable of dealing with relatively small datasets, and they present a wide range of configurations to generalize and

prevent overfitting. Machine learning models benefit from the increasing number of available data, which makes their performance constantly improve as time goes by."

Regarding point 3, we believe that the glacier geometry update component as well as its section in the manuscript should be kept. Indeed, the main novelty of the study and the model is its glacier-wide SMB component, but the fact that there is a glacier geometry update component makes it possible to simulate the evolution of glaciers at regional scale. We agree with the reviewer that the main focus of the study should be points 1 and 2, but we believe that point 3 can still be kept. With the changes presented here, we intend to give the main focus of the paper on the novel machine learning glacier-wide SMB approach, and mention the regional glacier evolution model as a platform in which this method has been implemented as a SMB component. It is important to present the combined SMB and glacier geometry update components, since this is a methodology paper, which will serve as a reference in the two future results papers. With the clarifications made in GC1 and GC2, and the fact that we specify that ALPGM is a regional model, we believe that the statement that ALPGM is a glacier evolution model is fair.

**2 Specific comments**

**Abstract L22**: for past and future climates.": remove future", since this has not yet been demonstrated.

The LOYO cross-validation evaluates the model's performance for time periods outside the data used for training. Since cross-validation can only be performed with past observed data, we believe that the current analysis from this study allows us to determine that the SMB modelling approach can be used in unseen (past or future) periods or climates. The only way to demonstrate that a model can be used for future climates is to validate it in the past over unseen periods (using LOYO), and to anticipate possible issues linked to hypotheses that might not be fulfilled.

P2 L6-16: although it is tempting to classify the models like this, I think that this list (and several other parts of the introduction) needs more precise definitions and a clearer positioning of the ALPGM model. In this list, you need to differentiate between the treatment of ice flow / glacier evolution by these models (on which your classification seems to be based, but not explicitly so) from the treatment of surface mass-balance (SMB), which is what your study is actually about. The "Physics-based models" that you list in fact often have no SMB module, and rely on external SMB as an external boundary condition in real-world applications. For the sake of clarity and given the scope of your study, I would rather focus on the hierarchy of SMB models (with SEB or even coupled Atmo-SEB models being the more advanced, and temperature index

models the simpler models). Please rethink this part of the introduction, as well as the following paragraph.

We agree that the current model classification is rather done based on glacier dynamics than SMB modelling. These two main components of glacier models have been identified in this section, but then they are mixed in the classification. Nonetheless, since ALPGM is a glacier evolution model and not just a SMB model, we believe that it is better to create two separate lists, one for SMB modelling and another one for glacier dynamics, instead of removing the glacier dynamics part which concerns the Δh parameterization used in ALPGM to update the glacier geometry.

Therefore, this section of the "Introduction" has been updated as follows:

"Glacier and hydro-glaciological models can help answer these questions, giving several possible outcomes depending on multiple climate scenarios. **(a) Surface mass balance (SMB) and (b) glacier dynamics both need to be modelled to understand glacier evolution on regional and sub-regional scales. Models of varying complexity exist for both processes.** In order to model these processes at large scale (i.e. on several glaciers at a catchment scale), some compromises need to be made, which can be approached in different ways:

**(a) Regarding SMB:**

> **1. Empirical models, like the temperature-index model (e.g. Hock, 2003), simulate glacier SMB through empirical relationships between air temperature and melt and snow accumulation.**

> **2. Statistical or machine learning models describe and predict glacier SMB based on statistical relationships found in data from a selection of topographical and climate predictors (e.g. Martin, 1974; Steiner et al., 2005).**

> **3. Physical and Surface Energy Balance (SEB) models take into account all energy exchanges between the glacier and the atmosphere, and can simulate the spatial and temporal variability of snowmelt and the changes in albedo (e.g. Gerbaux et al., 2005).**

**(b) Regarding glacier dynamics:**

> **1. Parameterized models do not explicitly resolve any physical processes, but implicitly take them into account using parameterizations, based on statistical or empirical relationships, in order to modify the glacier geometry. This type of models range from very simple statistical models (e.g. Carlson et al., 2014) to more complex ones based on different approaches, such as a calibrated equilibrium-line altitude (ELA) model (e.g. Zemp et al., 2006), a glacier retreat parameterization specific for glacier size groups (e.g. Huss and Hock, 2015) or volume/length-area scaling (e.g. Marzeion et al., 2012; Radic et al., 2014).**

> **2. Process-based models, like GloGEMflow (e.g. Zekollari et al., 2019) and OGGM (e.g. Maussion et al., 2019), approximate a number of glacier physical processes involved in ice flow dynamics using the shallow ice approximation.**

> **3. Physics-based models, like the finite elements Elmer/Ice model (e.g. Gagliardini et al., 2013), approach glacier dynamics by explicitly simulating**

> **physical processes and solving the full Stokes equations (e.g. Jouvet et al., 2009; Réveillet et al., 2015)..**

At the same time, **the use of** these different approaches strongly depend on available data, whose spatial and temporal resolutions have an important impact on the results' quality and uncertainties (e.g., Réveillet et al., 2018). **Parameterized glacier dynamics models and empirical and statistical SMB models** require a reference or training dataset to calibrate the relationships, which can then be used for projections with the hypothesis that relationships remain stationary **in space or in time**. On the contrary, process-based and specially physics-based **glacier dynamics and SMB models** have the advantage of representing physical processes, but they require larger datasets at higher spatial and temporal resolutions with a consequently higher computational cost (Réveillet et al., 2018). **For SMB modelling, meteorological** reanalyses provide an attractive alternative to sparse point observations, although their spatial resolution and suitability to complex high-mountain topography are often not good enough for high-resolution physics-based glacio-hydrological applications. However, parameterized models are much more flexible, equally dealing with fewer and coarser meteorological data as well as the state of the art reanalyses, which allows to work at resolutions much closer to glaciers' scale and to reduce uncertainties. The current resolution of climate projections is still too low to adequately drive most glacier physical processes, but the ever-growing datasets of historical data are paving the way for the training of parameterized machine learning models."
* * *
**P2 L34: Compared to other fields in geosciences": which ones?**

Mainly oceanography (e.g. Ducournau and Fablet, 2016; Lguensat et al., 2018) and climatology (e.g. Rasp et al., 2018; Jiang et al., 2018). The large datasets of ocean remote sensing data and climate reanalysis are being treated with approaches mixing deep learning, machine learning and physics. The scientific communities are much more advanced towards data-driven approaches, and have started to bridge the gap between data scientists and geophysicists, by adding physical knowledge into data science models.

These aspects have been added to the manuscript:

"Compared to other fields in geosciences**, such as oceanography (e.g., Ducournau and Fablet, 2016; Lguensat et al., 2018), climatology (e.g., Rasp et al., 2018; Jiang et al., 2018) and hydrology (e.g. Marçais and de Dreuzy, 2017; Shen, 2018),** we believe that the glaciological community has not yet exploited the full capabilities of these approaches."
* * *
**P2 L34: the glaciological community has remained quite oblivious to these advances": this is a subjective statement, you need to mention that this is your opinion.**

As suggested by both reviewers, this sentence has been rephrased in order to remove any negative or subjective connotations:

"… **we believe that the glaciological community has not yet exploited the full capabilities of these approaches**."

P3 L8: but all of them were linear, which are not necessarily the most suitable for modelling the nonlinear climate system": you have a very \statistical" view of linearity here. The 3 statistical models used in Maussion et al. are linear, yes, but they target individual SEB fluxes which are then transformed to be physical (e.g. by preventing negative precipitation or a non-closed SEB budget) and then used to compute the SMB. As a result, the full model M (as in SMB = M(y) with y the predictors and SMB the target variable) is nonlinear. This is important also in the context of traditional temperature index or degree day models, which can be compared to linear models applied to transformed predictors and as such, are also non-linear (e.g. by preventing melt for negative temperature or by transforming precipitation to solid precipitation). This is an important feature: without this non-linearity, they wouldn't work at all.

Indeed, here we refer to a statistical linearity. The example given by Fabien Maussion (with a temperature-index model being non-linear) would therefore be comparable to the simple multiple linear regression used in many studies, fed by PDDs and cumulative snowfall. By "pre-processing" the input predictors (daily temperatures to CPDD and precipitation to cumulative snowfall) one is introducing nonlinearities in the system. Nonetheless, this just creates new predictors which in this new space, can also behave linearly or nonlinearly. Indeed, the relationship between temperature and melt in the models you mention is nonlinear, but between PDDs and melt it is linear. So what we are referring to here, is that no matter how the input physical variables are pre-processed, their new relationships in the transformed space will probably be nonlinear as well. Therefore, there should be some benefits (as shown in this study) by switching to nonlinear models, even with the pre-processing of input predictors.

P4 L15: When most glacier models tend to incorporate more and more physical processes (Maussion et al., 2019; Zekollari et al., 2019), ALPGM takes an alternative approach based on data science." Are you talking about SMB or ice dynamics? Your data-science" is applied to the SMB problem here, and I believe it would be more appropriate to cite models of SEB/SMB in this sentence (e.g. Hock et al, Mölg et al, CROCUS, or similar).

This is related to the aspects raised in the P2 L6-16 comment. We are referring to both SMB and ice dynamics. For GloGEMflow, the addition of ice dynamics with the shallow ice approximation, and for OGGM, the ongoing developments on the SMB model (with the addition of shortwave radiation for instance). ALPGM does not use physics neither in SMB nor ice dynamics modelling. Therefore, the sentence has been updated as follows:

"When most glacier evolution models tend to incorporate more and more physical processes **in SMB or ice dynamics** (e.g., Maussion et al. (2019); Zekollari et al. (2019)), ALPGM takes an alternative approach based on data science **for SMB and parameterizations for glacier dynamics**."
* * *
**P5 L29-34: Although the features used as input (...) will likely have different biases." This paragraph seems out of context here and should be moved to the discussion**
* * *
Here we intended to make a quick introduction to the SMB modelling approach, and more specifically say that our approach aims at being a regional approach, which requires to be trained for each region of interest.

As suggested by the reviewer, this paragraph has been moved to the discussion section, in Sect. 4.2, at the end. Moreover, the paragraph from P5 L29-34 has been modified accordingly, to avoid any discussion items and just state the fact that the SMB modelling approach is regional:

"Annual glacier-wide SMBs are simulated using machine learning. **Due to the regional characteristics and specificities of topographical and climate data, this glacier-wide SMB modelling method is, for now, a regional approach.**"
* * *
**P6 L25: StatsModel" is spelled "StatsModels"**
* * *
This has been updated as suggested by the reviewer.
* * *
**P8 L1: The generated coefficients from the model serve to determine the significant predictors to be kept for the artificial neural network training." is Lasso part of the feature selection process of the ANN then? This raises interesting (and hard) questions concerning cross-validation and the model's real independence from training data. Furthermore, it gives an advantage to ANN over the linear models since their predictors are pre-filtered (see e.g. the double Lasso" method which makes this an advantage as well). Please comment.**
* * *
This sentence is in fact deprecated and it is now removed from the manuscript. This is also related to the comments in the last paragraph of GC1. During the development of the study, predictor selection via the Lasso was a hypothesis to be tested, but empirical tests using subsets of topo-climatic predictors as inputs of the ANN showed that it did not improve the results at all. The ANN is capable of choosing the relevant predictors by setting the weights of the connections of non-important predictors to zero. Moreover, as suggested by the reviewer, for the sake of equality in the comparison between statistical methods, the Lasso and ANN are fed with the same topo-climatic predictors.

The choice of input topo-climatic predictors is explained in Sect. 3.2.1; first with a literature review to target potential explanatory variables, and then with individual linear regressions to test the sensitivity of the SMB data to each individual predictor, similarly to what was done in Rabatel et al. (2016). This first choice of predictors is then used by each of the 3 statistical approaches: (1) All-possible multiple regressions tests the performance of all the possible subsets of predictors, (2) the Lasso performs a coefficient shrinkage to regularize the input predictors in order to discard them in a continuous way, and (3) the ANN gives specific weights to each connection of combined and non-combined input predictors at each neuron.

P9 L5-17: hyperparameters. As a non-specialist of "deep-learning", I need to ask: shouldn't this hyperparameter selection also be cross-validated? In Lasso, for example, the regularization parameter could be called an "hyper-parameter" and its selection takes place within the model tuning step, effectively making any external cross-validation (realized by LOGO and LOYO in your case) a true" out-of-sample validation. What about the ANN hyperparameters? Please comment.

Indeed, the hyperparameter selection needs to be done using cross-validation, but the process is quite different for Lasso than for an ANN.

For Lasso, there is only one hyperparameter (the α value), which is determined using cross-validation. This can be done with the Akaike Information Criterion (AIC), the Bayes Information Criterion (BIC) and a classical cross-validation with iterative fitting along a regularization path. This is explained in Sect. 2.2.3 "Lasso". Sci-kit learn provides different classes to help choose the best α value before fitting the Lasso model to data.

For the ANN, this becomes quite more complex. The list of hyperparameters to be fine-tuned is very long, so a brute force strategy of grid search or cross-validation of every single hyperparameter is strongly time consuming. There are smarter ways to proceed, especially with the knowledge of which types of optimizers, activation functions and hyperparameter values work best together. These aspects are discussed in Sect. 4.2. In an early stage, the ANN architecture with the number of neurons and layers, as well as the learning rate and type of optimizer, were cross-validated or tested for a few folds of the data. This early hyperparameter selection is not a proper cross-validation from A to Z, since in order to narrow down the range of values which gives the best performance, tests were first done in a few random folds. Then, once one has a better view of what can work for this dataset and architecture, a full cross-validation is performed with a few candidate hyperparameter values to choose the final ones. Therefore, in some ways it works similarly to Lasso and any machine learning training, where there is a first stage of hyperparameter tuning in order to choose a final configuration. Once the hyperparameters have been chosen, they remain constant throughout all the cross-validation, as it is done with Lasso. Due to the complexity of ANNs, the great number of hyperparameters and the fact that everything is open-source with new optimizers and approaches being released every month, the hyperparameter fine-tuning is a process that could be taken to infinity. One needs to know when to stop, when gains and additional tests stop bringing much added performance to the model.

These aspects have been added to the discussion in Sect. 4.2:

"In order to cope with the specific challenges related to each type of cross-validation, there are several hyperparameters that can be modified to adapt the ANN's behaviour. **Due to the long list of hyperparameters intervening in an ANN, it is not advisable to select them using brute force with a grid search or cross-validation. Instead, initial tests are performed in a subset of random folds to narrow down the range of best performing values, before moving to the full final cross-validations for the final hyperparameter selection.**"

Glacier geometry update: You call the geometry update a "parametrization" but in my opinion it isn't: you use an empirical Δh function perfectly known for each glacier since it is individually fitted. A true "parameterization" (like the one used in Huss and Hock 2015) would have the goal to work for any unseen glacier. Currently your model cannot be applied)(or validated) against unseen glaciers.

It depends on what we understand by "unseen glaciers". The Δh methodology used in Huss and Hock (2015), with a parameterization specific to 3 glacier sizes, requires to know at least the area of all the glaciers to be simulated, in order to choose a Δh function for them. In our case, in ALPGM, a glacier-specific Δh is computed for each glacier based on past DEMs. As explained in Sect. 3.3, these two DEMs cover the whole region of interest, therefore Δh functions are calculated for each glacier of the French Alps. With the SMB model trained from the 32 glaciers from the case study, the annual glacier-wide SMBs of all the glaciers in the French Alps can be simulated (the results will soon be presented in a separate paper), and since we have a specific Δh function for each glacier, their geometry can be updated with a better accuracy than using the 3 size-specific Δh functions used by Mathias Huss' studies. Therefore, ALPGM is capable of simulating the evolution of "unseen" glaciers (*i.e.* glaciers outside the 32 case study ones) in the same region.

Huss et al. (2008b), first presented the "Δh parameterization" for individual glaciers, as done in ALPGM, and they later presented the size-averaged functions in Huss et al. (2010). We believe that, despite being computed specifically for each glacier, the use of the word "parameterization" is fair in this context. The glacier-specific functions replace unresolved physical processes, and they rely on input data that is available for all glaciers. A parameterization refers to the procedure of replacing complex (geophysical in this case) processes by simplified processes. In the words of Stensrud (2007), "Parameterization schemes": "There are always physical processes and scales of motion that cannot be represented by a numerical model, regardless of the resolution […]. Parameterization is the process by which the important physical processes that cannot be resolved directly by a numerical model are represented".

Figure 4: Since you have DEMs (and geodetic MBs) from all blue glaciers in Figure 4, can you apply your model to them as well and compare? This would be a good (but partial) out-of-sample validation (partial" because you still need knowledge about the glacier's Δh).

The results of the proposed glacier-wide SMB reconstruction methodology of this paper applied to all the glaciers in the French Alps will soon be presented in a separate paper. Indeed, it would be a complementary way to cross-validate the glacier-wide SMB model, but the extension to other glaciers outside the ones from the case study is out of the scope of this (already quite long) paper. Moreover, comparing simulated annual glacier-wide SMBs to cumulative goedetical SMBs is quite limited, as it only serves to determine the bias of the model, since simulations would have to be summed in order to produce the interannual glacier-wide SMB, with positive and negative errors potentially being compensated or cancelled among them.

> Glacier ice thickness: To avoid confusion: if still applicable after revision, mention here that ice-thickness are only used for the 2003-2016 test run, and not for the rest of the model workflow.

This aspect has been addressed together with the next comment. Please see the updated Sect. 3.1.2 from the next comment.

> Glacier topographical variables: from an email question to the authors I know that the topographical predictors (e.g. area, slope) are time-dependant and obtained from various DEM snapshots. This needs to be explained here. Regardless of this missing explanation, this raises questions about the overall applicability of the method to unseen situations (see general comment).

Sect. 3.1.2 has been updated in order to include this information:

"**The topographical data used for the training of the glacier-wide SMB machine learning models is taken from the multitemporal inventory of the French Alps glaciers (e.g. Gardent et al. (2014)) partly available through the GLIMS Glacier Database (NSIDC (2005)). We worked with the 1967, 1985, 2003 and 2015 inventories (Gardent et al. (2014), with 2015 update). Between these dates, the topographical predictors are linearly interpolated. On the other hand, in the glacier evolution component of ALPGM (Fig.1, step 5), the topographical data are re-computed every year for each glacier from the evolving and annually updated glacier-specific ice thickness and DEM rasters (Sect. 3.1.3). For the computation of the glacier-specific geometry update functions, two DEMs covering the whole French Alps have been used:** (1) one from 2011 generated from SPOT5 stereo-pair images, acquired on 15 October 2011; and (2) a 1979 aerial photogrammetric DEM from the French National Geographic Institute (Institut Géographique National, IGN), processed from aerial photographs taken around 1979. Both DEMs have an accuracy between 1 and 4 meters (Rabatel et al. (2016)), and their uncertainties are negligible compared to many other parameters in this study."

> P16 L2: For the training of the ANN, no combination of topo-climatic features is done as previously mentioned". I have a hard time finding where this is explained. Is this the part with Lasso? In any case, the predictor selection for ANN needs to be explained here for consistency and to help the reader.

The fact that no combination of topo-climatic predictors is done for the ANN (unlike for Lasso as shown in Eq. 4) is explained in Sect. 2.2.4 "Deep artificial neural network", lines 8-10. As explained in our response from comment P8 L1, no predictor selection is done for the ANN. Each of the 3 machine learning algorithms performs its own predictor selection.

In order to aid the reader, this information has been added again in a brief way in this section:

"For the training of the ANN, no combination of topo-climatic predictors is done as previously mentioned **(Sect. 2.2.4), since it is already done internally by the ANN**."

P16 L9: Latitude and longitude seem to play an important role when combined with snowfall.". I don't really understand the climatological explanation that follows this statement. If the reanalysis data is accurate, then these east-west and north-south differences should already be in the training data. If anything, these combination of predictors play the role of bias correction - or are the result of luck (which is often the case with many co-linear predictors).

Meteorological reanalyses are not capable of perfectly reproducing all the complex precipitation patterns found in mountainous regions. As explained in Vionnet et al. (2016), the SAFRAN reanalysis is more uncertain, and likely negatively biased at the higher altitudes in the French Alps. Because of the distribution of high altitude massifs towards the northern and eastern sides of the French Alps, this precipitation bias is more present in the northern massifs (see Table 6 from Vionnet et al. 2016 included in this reply) and the eastern massifs.

TABLE 6. Error statistics (bias and STDE; m) from the comparison between measured and simulated snow depth using AROME-SC and SAFRAN-SC over the French Alps and subregions for winters from 2010/11 to 2013/14. The location of the subregions is shown in Fig. 1 and the number of stations is specified in Table 1.

| | AROME-SC | | SAFRAN-SC | |
|---|---|---|---|---|
| | Bias | STDE | Bias | STDE |
| North | 0.51 | 0.46 | 0.23 | 0.38 |
| Central | 0.50 | 0.49 | 0.21 | 0.36 |
| South | 0.25 | 0.51 | 0.15 | 0.37 |
| Extreme south | 0.08 | 0.31 | −0.01 | 0.29 |
| French Alps | 0.40 | 0.50 | 0.18 | 0.37 |

We believe, as suggested by the reviewer, that these two topographical parameters play the role of bias correction. Since the machine learning models learn from past data, they can find relationships between precipitation and regions, compensating some of the bias found in the climate data. A quick analysis of the coefficients from the Lasso causal inference analysis revealed that the latitude and longitude predictors modulate the glacier-wide SMB signal in the French Alps, with a positive northeastern gradient. Therefore, these two predictors likely correct the underestimation of precipitation for the northeastern massifs of the French Alps. In the original Lasso causal inference analysis, where the combination of topo-climatic predictors was taken into account, it was shown how longitude*winter snowfall and latitude*winter snowfall were among the most important predictors. This further assesses the importance of latitude and longitude to correct the bias of snowfall in certain regions.

This information has been added in Sect. 3.2.2 "Causal analysis":

"**In a second term, topographical predictors do play a role, albeit a secondary one. The slope of the 20% lowermost altitudinal range, the glacier area, the glacier mean altitude and aspect help to modulate the glacier-wide SMB signal, which unlike point or altitude-**

**dependent SMB, partially depends on glacier topography (Huss et al., 2012). Moreover, latitude and longitude are among the most relevant topographical predictors, which for this case study are likely to be used as bias correctors of precipitation of the SAFRAN climate reanalysis. SAFRAN is suspected of having a precipitation bias, with higher uncertainties for high altitude precipitation (Vionnet et al., 2016). Since the French Alps present an altitudinal gradient, with higher altitudes towards the eastern and the northern massifs, we found that the coefficients linked to latitude and longitude enhanced glacier-wide SMBs with a north-east gradient.**"

Lon/Lat Predictors: this is a subjective opinion, but I suggest to remove Lon/Lat predictors from the set. They should not explain anything which isn't in the climate and topographical predictors already, and using lon/lat seriously hinders the applicability of the model to larger areas.

We agree with the reviewer on that point, provided perfect, unbiased climate reanalysis were available. However, as explained in the previous comment, the latitude and longitude topographical predictors play in our case a role in precipitation adjustment/bias correction. The use of these predictors does not hinder the applicability of this approach, since as mentioned in Sect. 3.2, this glacier-wide SMB modelling approach is regional, and the training is specific for each region (determined by the spatial coverage of climate forcings and their quality, the sensitivity of SMB to climate and topographical data). If this SMB modelling approach was to be applied for instance to the whole European Alps, it would require first to switch to climate forcings with a full coverage of the European Alps, add as much SMB data of glaciers in the region as possible and then to retrain the model.

ALPGM should not be mistaken for a global glacier evolution model. The way the SMB machine learning models have been trained so far, is region-based. In the future, with an ambitious training with lots of data, perhaps it would be possible to make the SMB model more global. But it would require re-thinking many things, in order to consider different SMB-climate sensitivities.

As discussed in GC3, the manuscript has been updated, referring to ALPGM as a regional model throughout the different sections.

Figs 6 and 7: although visually appealing, the use of different colorscales for the ANN and linear models is misleading. All four plots are exact same and should have the same colorscale, min-max range, x-y axis, etc.

We agree that the min-max ranges should be the same. However, the fact that linear methods are in purple-red and nonlinear in blue-green is not random. This colour code is respected throughout the paper (Figs. 6, 7, 8, 9 and 10), with all figures referring to the different models. With the same min-max range, since the colour gradient is extremely similar, it is very easy to compare linear and nonlinear plots while keeping the color code. It is worth mentioning that keeping the min-max range constant has improved the figures, allowing them to show how for LOYO, deep learning performs slightly worse than for LOGO.

The plots have been updated, changing the font size of the axis and keeping a constant min-max range for all the scatter plots.

Figs 6 and 7 and corresponding discussion about explained variance: a possible improvement to describe the models errors is to plot binned model error (residuals) as a function of the target variable (here, SMB), or use a Q-Q Plot. It would display in a more quantitative way the non-normal distribution of model errors (visible on the scatter plots by a flattening on both ends of the scatter), further making your point that ANN is better (but not perfect) at reproducing the true variance of the data.

As suggested by the reviewer, we have added an extra plot with the error distribution of the deep learning glacier-wide SMB model, in the temporal and spatial dimensions. The effect explained in these sections shows up, with higher errors for extreme values, mostly due to their underrepresentation in the dataset, being outliers. This new plot has been added in the supplementary material and it has been added as a reference during the discussion involving Fig. 6 and 7.

[Figure]

**Figure S7:** Error distribution of deep learning (without weights) glacier-wide SMB simulations for the 1984-2015 period for the 32 case study glaciers. (a) Performance in the spatial dimension using LOGO cross-validation; (b) performance in the temporal dimension using LOYO cross-validation. The red line corresponds to a $5^{th}$ order polynomial fit.

Figure 10: a striking feature of figure 10 and not discussed in the manuscript is the clear tendency of LASSO to overestimate MB in the second half of the period and underestimate MB in the first half. I guess it is a result of more frequent negative MBs in the second half, which are underestimated by the model with an obvious lower variance, but is this the only reason?

This aspect has been briefly discussed in the last sentence in Sect. 3.2.4 "Temporal predictive analysis". The lack of complexity of the linear Lasso is more obvious in the more complex temporal dimension compared to the spatial dimension. As explained in Sect. 4.2 "Training deep learning models with spatiotemporal data", the interannual variability in the glacier-wide SMB signal (temporal dimension) is controlled by climate,

and the topography (spatial dimension) is responsible for the modulation of the signal. This lack of complexity of the linear model towards the temporal glacier-wide SMB signal, means that the model is underfitting the data, and by trying to minimize the overall error (increasing the variance), it introduces a different (negative for the first half and positive for the second) bias for the two clearly distinct periods of glacier-wide SMB. This means that the Lasso model manages to have an acceptable RMSE at the price of introducing an important bias.

[Figure]

This explanation has been extended to address these details:

"The important bias present only with Lasso is representative of its lack of complexity towards nonlinear structures, **which results in an underfitting of the data. The average error is not bad, but it shows a high negative bias for the first half of the period, which mostly has slightly negative glacier-wide SMBs, and a high positive bias for the second half of the period, which mostly has very negative glacier-wide SMB values**."

Glacier geometry evolution validation: these results are not too surprising. Since your evolution model knows exactly where mass is going to be removed (based on data going up to 2011). This test is basically a bias test: if the model has no bias, ice is going to be removed at the right place (because you know where to remove it) and the area will be correct, provided that the ice thicknesses are more or less accurate.

Indeed, it can be seen as a bias test since the errors are accumulated throughout the years, and the final result is the one which is observed and compared. Nonetheless, it serves to test that the glacier evolution model (SMB component + glacier geometry update + ice thickness data) is capable of reproducing past glacier evolution. Since the glacier-wide SMB data used here is cross-validated, this test is representative of the model's end-to-end performance. More details regarding the uncertainties and the purpose of Fig. 11 are discussed in the next question.

P24 L1: Even for a 12-year period, the initial ice thickness remains the largest uncertainty": this statement is not supported by your results, since this is the only uncertainty you consider in Fig. 11. Here, you could add model uncertainty by using out-of-sample validation (by training the model with data only before 2003 and using LOGO), or use uncertainty measures derived by cross-validation. The issue with the h method raised above would remain, though.

Indeed, the fact that we said that the main uncertainty came from the initial ice thickness data used was purely based on similar studies (e.g. Huss and Hock, 2015) which claimed this. In order to verify this by ourselves, we have added the uncertainty linked to the Δh glacier geometry function calculation in Fig. 11. From the automatic process to compute it from the DEM difference, we estimated an average uncertainty of ±10% based on the polynomial fits over the altitude difference values (Fig. S4). By adding this uncertainty, we can now claim, as stated by other studies, that the uncertainties linked to the initial ice thickness are greater than the uncertainties linked to the glacier geometry update parameterization.

Regarding the out of sample validation proposed here, we believe that training the model with only data before 2003 would not be representative of the model's real performance, since we would be using only half of the data available, thus severely penalizing the model. Instead, we have used the LOGO models, which allow a true out of sample validation for each glacier. These results have been included in Fig. 11 as well. Therefore, now the test of Fig. 11 allows to show the end-to-end glacier evolution performance of ALPGM for the 32 case study glaciers, including a true out of sample glacier-wide SMB values, the uncertainties linked to the initial ice thickness data and the uncertainties linked to the glacier geometry update functions. Indeed, there is no way to truly cross-validate these geometry update functions, since they need to be calibrated for the longest available period (1979-2011), but we believe that this test still allows to quantify and prove the glacier evolution simulations' performance.

The explanations in Sect. 3.3 "Glacier geometry evolution: Validation and results" have been updated accordingly, including Fig. 11:

"In order to evaluate the performance of the parameterized glacier dynamics of ALPGM, **coupled with the glacier-wide SMB component,** we compared the simulated glacier area of the 32 studied glaciers with the observed area in 2015 from the most up-to-date glacier inventory in the French Alps. Simulations were started in 2003, for which we used the F19 ice thickness dataset. In order to take into account the ice thickness uncertainties, we ran three simulations with different versions of the initial ice thickness: the original data, -30% and +30% of the original ice thickness in agreement with the uncertainty estimated by the authors. **Moreover, in order to take into account the uncertainties in the Δh glacier geometry update function computation, we added a ±10% variation in the parameterized functions** (Fig. 11).

Overall, the results illustrated in Fig. 11 show a good agreement with the observations. Even for a 12-year period, the initial ice thickness remains the largest uncertainty, with almost all glaciers falling within the observed area when taking it into account. The mean error in simulated surface area was of **10.7%** with the original F19 ice thickness dataset. Other studies using the Δh parameterization already proved that the initial ice thickness is the most important

uncertainty in glacier evolution simulations, together with the choice of a GCM for future projections (Huss and Hock, 2015)."

[Figure]

**Figure 11**. Simulated glacier areas for the 2003-2015 period for the 32 study glaciers using a deep learning SMB model without weights. Squares indicate the different F19 initial ice thicknesses used taking into account their uncertainties and triangles the uncertainties linked to the glacier-specific geometry update functions. For better visualisation, the figure is split in two with the two largest French glaciers on the right.

P25 L26: we trained an ANN only with monthly average temperature and snowfall, without any topographical predictors". These experiments should become the central component of your study, not the other way around (see general comment).

This aspect is discussed in detail in our replies to GC1 and GC2. The good performance of a glacier-wide SMB model without topographical predictors, shows again the secondary role that they play. As discussed in Huss et al. (2012) "Conventional versus reference-surface mass balance", and it the comments in The Cryosphere between Leclrercq et al. and Huss et al., there is a debate regarding their importance, but from a statistical point of view based on the data used in this study, we can state that in our case they do play a secondary role.

Authors reply to Anonymous Referee #1's review on "Deep learning applied to glacier evolution modelling"

**Anonymous Referee #1**

The research presented in this manuscript shows promising results in the application of an ANN model used for surface mass balance modelling. The manuscript is, for the most part, well organized. The manuscript can be greatly improved by increasing clarity and specificity throughout. I hope that my comments are helpful to the authors in this effort. No single one of my comments identifies a major flaw with the manuscript; rather, there are many small changes that I believe can be made to improve the quality of the paper. I have organized my comments in sequential order by section, preceded by one general note.

We would like to thank the reviewer for the time dedicated to read the manuscript and for the overall positive feedback. All the detailed comments will hopefully improve the overall clarity and fluidity of the manuscript. All points raised during the review have been addressed and answered, in the following detailed sections, and the manuscript has been updated accordingly.

Small changes within paragraphs are shown in bold, in order to distinguish them from their context.

**General Note:**

The difference between "machine learning" and "deep learning" is not clearly defined in the literature, but a 6-layer ANN is likely at the very tip of what may constitute "deep learning". Considering that, as you note, deep learning is not a common tool among the glaciological community, it would be good to provide further context as to what an ANN is (a type of model, which is relatively simple in the deep learning world as compared to, say, a convolutional neural network or long short-term memory network) versus what deep learning is. I believe that this is required especially because the title refers to deep learning broadly, not a deep ANN specifically, and it should be made clear that there is much more to deep learning than ANNs.

Indeed, the jargon in the machine and deep learning fields is often not well defined. Nonetheless, deep learning is a subfield of machine learning, involving ANNs with more than one hidden layer. Therefore, one could determine the following hierarchy between these concepts:

Authors reply to Anonymous Referee #1's review on "Deep learning applied to glacier evolution modelling"

[Figure]

ANNs are an example of machine learning, and within ANNs one needs to choose an architecture: single or deep (multiple) hidden layers, and a type of ANN: feedforward (used in our study), convolutional, LSTM.

The title refers to deep learning, which is broad in the sense that it could imply the use of different types of ANN. But in Sect. "2.2.4 Deep artificial neural network" line 16, we specify that a feedforward fully-connected ANN is used. We do not understand what the reviewer means by "and it should be made clear that there is much more to deep learning than ANNs", since deep learning is a subfield of machine learning constituted only by ANNs with multiple hidden layers.

In order to increase clarity for the reader, we have specifically mentioned this aspect in Sect. 2.2.4, in lines 16-22, which now reads as:

"Artificial neural networks (ANNs) are nonlinear statistical models inspired by biological neural networks (Fausett (1994); Hastie et al. (2009)). A neural network is characterized by: (1) the architecture or pattern of connections between units and the number of layers (input, output and hidden layers); (2) the optimizer: the method for determining the weights of the connections between units; and (3) its (normally nonlinear) activation functions (Fausett (1994)). **When ANNs have more than one hidden layer (*e.g.* Fig. 3), they are referred to as deep ANNs or deep learning**. The description of neural networks is beyond the scope of this study, so for more details and a full explanation please refer to Fausett (1994), Hastie et al. (2009), as well as Steiner et al. (2005, 2008) where the reader can find a thorough introduction to the use of ANNs in glaciology."

**Sequential notes:**

Page 1, Line 22: What does "individual glaciers at regional scale" mean? Do you mean to say, "individual glaciers within the same region"?

We mean the reconstruction of SMB series of individual glaciers for a whole region.

The sentence has been rephrased to improve its clarity:

"… that can serve to reconstruct or simulate SMB time series for individual glaciers in a whole region for past and future climates."

1 Introduction: Page 1, Line 25: "...being climate proxies which can clearly depict the evolution of climate for the global audience"; remove "clearly", if the evolution of climate was clear for the global audience, then why is there so much disagreement among the global audience?

The sentence has been adapted as suggested by the reviewer.

Page 1, Line 26: "For the coming decades..."; I believe this should be "In the coming decades..."

The sentence has been adapted as suggested by the reviewer.

Page 1, Line 28: "The reduction in ice volume may produce an array of consequences which requires to be properly predicted." This sentence, and the following, is vague. What consequences are you talking about? Be explicit.

The sentence has been rephrased as it follows to specify the consequences and importance of glacier retreat:

"**The reduction in ice volume may produce an array of hydrological, ecological and economic consequences in mountain regions which requires to be properly predicted**. These consequences will strongly depend on the future climatic scenarios, which will determine the timing and magnitude for the transition of hydrological regimes (Huss and Hock (2018)). **Understanding these future transitions is key for societies to adapt to future hydrological and climate configurations.**"
* * *
Page 2, Line 2: "For any glacier model..."; Saying "any" makes this sentence too broad and not necessarily true. Be explicit for the classes/types/purposes of modes which require SMB and glacier dynamics (e.g. "SMB and glacier dynamics both need to be modelled to understand glacier evolution on regional and sub-regional scales. Models of varying complexity exist for both processes.")

The sentence has been rephrased as suggested by the reviewer.
* * *
Page 2, Line 18: "...these different approaches strongly depend on available data..."; Change to "...the use of these different approaches strongly depend on available data..." since it the model usage, not the model itself, which depends on what data one has.

The sentence has been rephrased as suggested by the reviewer.
* * *
Page 2, Line 21: "...relationships remain stationary."; Change to "...relationships remain stationary in time."

The sentence has been rephrased as follows, including as well the spatial dimension:

"… which can then be used for projections with the hypothesis that relationships remain stationary in time."
* * *
Page 2, Line 34: "...the glaciological community has remained quite oblivious to these advances..."; Oblivious is a strongly negative word to use here, and it is a disservice to insult your readers.

We agree that this word choice is not the most suitable in this context, because of its negative and subjective connotations. The sentence has been rephrased as follows:

"… **we believe that the glaciological community has not yet exploited the full capabilities of these approaches**."
* * *
Page 6, Line 10: "...relevant predictors must be selected, performing a sensitivity study..."; Change to "...relevant predictors must be selected, so we perform a sensitivity study..."

The sentence has been rephrased as suggested by the reviewer.
* * *
Page 6, Line 14 and Equation 1: Is there a reference for this "effective way of expanding the training dataset"?

This is a common practice in regression, similarly to data augmentation and what an ANN does internally combining the input parameters in each hidden layer. It must of course be done before subset selection or regularization. It is explained in Weisberg (2014), Sect. 10.2, which has been added as a reference for this sentence.
* * *
Page 7, last sentence: Here you describe the types of cross validations available in ALPGM. Which did you use?

We use the cross-validation with iterative fitting along a regularization path. This has been now specified after the sentence:

"ALPGM performs different types of cross-validations to choose from: the Akaike Information Criterion (AIC), the Bayes Information Criterion (BIC) and a classical cross-validation with iterative fitting along a regularization path **(used in the case study)**."
* * *
Page 8, Line 6: "... (2) the optimizer: the method for..."; change to "...(2) the optimizer, which is the method for..."

The sentence has been rephrased as suggested by the reviewer.
* * *
Page 8, Line 6: "...(3) its (possibly nonlinear) activation functions..."; When are activation functions linear?

They are almost never linear, but it is still a possibility. For specific cases where one does not want to restrict the output values within a certain range, using a linear activation function allows to produce real values. Nonetheless, using them in more than one layer in a deep ANN does not make any sense.

The sentence has been adapted as follows:

"(3) its (**usually** nonlinear) activation functions"
* * *
Page 8, Line 10: "...allowing to train deep neural networks..."; change to "allowing the training of deep neural networks..."

The sentence has been rephrased as suggested by the reviewer.
* * *
Page 8, Line 11: "...ANNs are best suited when the quality of predictions prevails over the interpretability of the model." This is vague, and does not help readers know when ANNs are 'best suited'. How are either of these things quantified?

This cannot be strictly quantified, it depends on each field and situation. It is based on the understanding of the process which is being modelled. If a certain process is well understood, and the variables which are involved are well known, then it is acceptable to focus on prediction rather than causality. One can build a prototype model with the previous knowledge of which variables are meaningful.

The sentence has been rephrased as follows in order to clarify the sentence with respect to the goal of this study:

"As their learnt parameters are difficult to interpret, **ANN are adequate tools when the quality of predictions prevails over the interpretability of the model (the latter likely involving causal inference, sensitivity testing or modelling of ancillary variables). This is precisely**

**the case in our study context here, where abundant knowledge about glacier physics further helps choosing adequate variables as input to deep learning**"
* * *
Page 10, Line 21: "...should be long enough to be representative of the glacier evolution..."; How long is 'representative'? Representative of what? How does one know this?

This sentence is directly related to what has been stated previously in the same paragraph. The time difference between the two DEMs depends on the achievable signal-to-noise ratio, meaning that if a glacier is losing mass at a high pace, one will be able to use a shorter time period between the two DEMs. This is of course done with the hypothesis of glacier shrinkage in the future due to climate change, so in order to have a representative parameterization of how the glacier retreats, we need to find a period of glacier retreat in the recent past.

Due to the confusion produced by this sentence, and the fact that the necessary information is already conveyed in the same paragraph, we removed this sentence:

"As discussed in Vincent et al. (2014), the time period between the two DEMs used to calibrate the method needs to be long enough to show important ice thickness differences. The criteria will of course depend on each glacier and each period, but it will always be related to the achievable signal-to-noise ratio. Vincent et al. (2014) concluded that for their study on the Mer de Glace glacier (28.8 km2, mean altitude = 2868 m.a.s.l.) in the French Alps, the 2003-2008 period was too short, due to the delayed response of glacier geometry to a change in surface mass balance. Indeed, the results for that 5-year period diverged from the results from longer periods. "
* * *
Page 11, Line 3: Refer to Figure 4 here

A reference to Figure 4 has been added.
* * *
Page 11, Line 5: Is there a reference to this study?

This study is the main author's (Jordi Bolibar) PhD project, and since my PhD thesis manuscript is still to be written there is not an available reference yet.
* * *
Page 11, Line 9: "...using remote sensing based on changes in glacier volume and the snow line altitude is used..."; Remove second "is used"

The "is used" part should not be removed, otherwise the sentence would be left with a subject without verb.

"SUBJECT (An annual glacier-wide SMB dataset reconstructed using remote sensing based on changes in glacier volume and the snow line altitude) + VERB (is used)"

Commas have been added to give pause and increase clarity:

Authors reply to Anonymous Referee #1's review on "Deep learning applied to glacier evolution modelling"

**"An annual glacier-wide SMB dataset, reconstructed using remote sensing based on changes in glacier volume and the snow line altitude, is used"**
* * *
Page 12, Figure 4: Axes should be labelled

In order to keep Fig.4 more compact in a single column, the coordinates of the bottom left corner and the top right corner have been added in the legend to guide the reader. We believe this information should be enough to properly read the map.
* * *
Page 13, Line 2: Cite RGI (check here for reference: https://www.glims.org/RGI/)

The RGI Consortium (2017) reference has been added as follows:

RGI Consortium (2017): Randolph Glacier Inventory(RGI) – A Dataset of Global Glacier Outlines: Version 6.0. Technical Report, Global Land Ice Measurements from Space, Boulder, Colorado, USA. Digital Media. Doi: 10.7265/N5-RGI-60
* * *
Page 13, Line 12: Qualifications here are vague (e.g. "quite satisfactorily", "good over-all", "certain altitudinal ranges"). Give quantitative measures of "goodness", and refer to specific parts of Figure S2 that demonstrate what you're talking about.

The paragraph has been rephrased to improve the precision of the statements with references to Fig. S2:

"The simulated ice thicknesses for Saint-Sorlin (2 km$^2$, mean altitude = 2920 m.a.s.l., Écrins cluster) and Mer de Glace (28 km$^2$, mean altitude = 2890 m.a.s.l., Mont-Blanc cluster) glaciers are satisfactorily modelled by F19. **Mer de Glace's tongue presents local errors of about 50 m, peaking at 100 m (30% error) around 2000-2100 m.a.s.l, but the overall distribution of the ice is well represented. Saint Sorlin glacier follows a similar pattern, with maximum errors of around 20 m (20% error) at 2900 m.a.s.l. and a good representation of the ice distribution.**"
* * *
Page 13, Line 26: This sentence can be improved by maintaining consistency across clause structure. You use "we verb" statements (e.g. we go through, we assess, and we show) for all clauses except for "the building of the machine learning SMB models".

The sentence has been rephrased in order to keep it more consistent:

"In this section, we go through the selection of SMB predictors, **we introduce the procedure for building machine learning SMB models**, we assess their performance in space and time and we show some results of simulations using the French alpine glaciers dataset."

Page 14, Line 25 (and paragraph): You discuss that you dynamically calculate the accumulation/ablation periods based on the CPDD, and that you keep constant periods to account for winter and summer snowfalls. Later, you use 'transition months' as predictors – are these predictors kept constant, or dynamically calculated? Are results improved when the transition months are dynamically computed? I ask because I would expect that what constitutes a 'transition month' may change in the future. Or do you think that this approach, applied to more variables, then forces the model to depend too much on CPDD when the CPDD is not the only variable involved in melt?

This dynamical separation between ablation and accumulation periods is done to compute the seasonal meteorological data: the CPDD (temperature in ablation season), the winter snowfall and the summer snowfall. These three variables are introduced as climate predictors in Eq. 3. However, there are as well the monthly temperature and snowfall values in Eq. 3, which in Sect. "3.2.2 Causal analysis" are sometimes referred as transition months. The machine learning models receive all the monthly data as part of Eq. 4 and then determine which months are more relevant to explain the glacier-wide SMB of glaciers in this region. The fact that some transition months (between the ablation and accumulation periods) showed up as relevant predictors in the causal analysis, is purely based on the relationships found in the meteorological data between 1959 and 2015 for some glaciers, and between 1984 and 2015 for most glaciers of the dataset. As explained in Sect. "1 Introduction", lines 20-21, parameterized and statistical models work with the hypothesis that the relationships found in data remain stationary in time. This is of course not totally true in our case, which is why we decided to dynamically compute the ablation season (CPDD) to account for the (likely) longer ablation periods in the future. Therefore, the seasonal meteorological data adapts to future climate changes, but the individual relationships found in monthly data remain constant. Nonetheless, since there are many predictors for monthly data, their importance is very distributed, so these stationary relationships based on past climate data should not have such an important effect.

Page 15, Equations 2 and 3: Are input variables normalized? If so, how?

The input variables are only normalized for the Lasso. The ANN includes batch normalization internally, so raw data is fed directly to the input layer. This is already mentioned in "2.2.4 Deep artificial neural network", but it was indeed not specified for the Lasso. Therefore, a new line has been added in "2.2.3 Lasso" as follows:

"All input data is normalized by removing the mean and scaling to unit variance."

Page 15, Line 20: When you say 'linear machine learning', are you referring to the linear regression methods? Be consistent in how you refer to your methods.

With "linear machine learning" we mean the linear methods used in this paper (OLS and Lasso). "Linear regression methods" and "linear machine learning" are equivalent

terms in this paper, since we are only working with regression. The term "linear regression" is only used as "multiple linear regression" when referring to OLS or stepwise multiple linear regression. The terms "linear machine learning" and "nonlinear machine learning" are the ones used throughout the paper, especially in Sect. 3 and 4 to refer to the differences found between linear methods and nonlinear deep learning.

In order to avoid confusion, the sentence has been changed as follows using the plural to refer to both linear machine learning models:

"For the linear machine learning model**s** training, we chose a function f that …"

**Page 15, Line 20: How did you choose the function f?**

Function f is based on the data expansion mentioned in Page 6, line 14. The idea is to linearly combine topographical and seasonal climatic data, with the exception of the monthly data. Monthly data is not combined to avoid the generation of an unnecessary number of predictors. The sentence has been adapted as follows for clarity:

"For the linear machine learning models training, we chose a function $f$ that linearly combines $\hat{\Omega}$ and $\hat{C}$, generating new combined predictors (Eq. 4**). In $\hat{C}$, only ${\Delta CPDD}$, ${\Delta WS}$, and ${\Delta SS}$ are combined, to avoid generating an unnecessary amount of predictors with the combination of $\hat{\Omega}$ with ${\Delta \overline{T}_{\operatorname{mon}}}$ and ${\Delta \overline{S}_{\operatorname{mon}}}$.**"

LaTex print:

For the linear machine learning models training, we chose a function $f$ that linearly combines $\hat{\Omega}$ and $\hat{C}$, generating new combined predictors (Eq. 4). In $\hat{C}$, only $\Delta CPDD$, $\Delta WS$, and $\Delta SS$ are combined, to avoid generating an unnecessary amount of predictors with the combination of $\hat{\Omega}$ with $\Delta \overline{T}_{\mathrm{mon}}$ and $\Delta \overline{S}_{\mathrm{mon}}$.

**Page 15, Equation 25: You create linear models using the predictors shown here. You then create nonlinear models using only the predictors in Equations 2 and 3. Then, you compare the results of these models and conclude that the nonlinear model is better because of the nonlinear nature of the model; however, how do you know that the improved performance is not simply due to using a different set of predictor variables? Your argument would be more convincing if you first showed that the linear model performance improved when you change predictor variables from the standard case (those only in Equations 2 and 3) to the combination case (Equation 4), and then showed that a nonlinear model using variables from the standard case outperformed even this improved linear model.**

For both OLS and Lasso, a subset selection or coefficient shrinkage is done in order to reduce the number of kept predictors. Even in the expanded Eq. 4, the original predictors from Eq. 2 and 3 are still there, so they are potential candidates to be chosen. We believe that linear models trained with Eq. 4 can be compared to a deep ANN trained with Eq. 2 + 3, because as mentioned in Page 16, line 2, the ANN already performs all the possible combinations in each layer by combining all the input predictors. For OLS, it would not change anything to test only using Eq. 2 + 3, since we are computing all the possible combinations of Eq. 4 already, so the Eq. 2 + 3

subset is already covered in the current case. For Lasso, the situation would be slightly similar. Some early tests were already done without combination of climate and topographical predictors with worse results. Moreover, as Fig. 5 shows, the top predictor and many of the top predictors in Lasso are combined, so the added value is already shown in the results.

In order to clarify these aspects, a sentence has been added in Sect. 3.2.1 as follows:

"Early Lasso tests (not shown here) using only the predictors from Eq. 2 and 3 demonstrated the benefits of expanding the number of predictors, as it is later shown in Fig. 5."

**Page 15, Equation 5: Is there a missing '('? This equation ends with ')_g,y'**

Indeed, there is a format problem with the parenthesis. Eq. 4. This has been fixed as suggested by the reviewer.

**Page 16, Line 1: It is not clear to me why there are 50 predictors, when there are 33 coefficients in Equation 4.**

In fact, this is a mistake. There used to be 50 predictors, but after a small change in the code they should have been updated to 55. From the 33 predictors, two are the mean monthly temperature and the monthly snowfall, which account for 12 predictors each (one value per month). Therefore: $33 - 2 + 24 = 55$ predictors.

This number has been fixed throughout the manuscript and a sentence has been added to clarify this aspect:

"32 glaciers over variable periods between 31 and 57 years result in 1048 glacier-wide SMB ground truth values. **For each glacier-wide SMB value, 55 predictors were produced following Eq. 4: 33 combined predictors, with ${\Delta \overline{T}_{\operatorname{mon}}}$ and ${\Delta \overline{S}_{\operatorname{mon}}}$ accounting for 12 predictors each, one for each month of the year.** All these values combined produce a 1048x55 matrix, given as input data to the OLS and Lasso machine learning libraries."

LaTex print:

32 glaciers over variable periods between 31 and 57 years result in 1048 glacier-wide SMB ground truth values. For each glacier-wide SMB value, 55 predictors were produced following Eq. 4: 33 combined predictors, with $\Delta \overline{T}_{\operatorname{mon}}$ and $\Delta \overline{S}_{\operatorname{mon}}$ accounting for 12 predictors each, one for each month of the year. All these values combined produce a 1048x55 matrix, given as input data to the OLS and Lasso machine learning libraries. Early Lasso tests (not shown here) using only the predictors from Eq. 2 and 3 demonstrated the benefits of expanding the number of predictors, as it is later shown in Fig. 5. For the training of the ANN, no combination of topo-climatic predictors is done as previously mentioned (Sect. 2.2.4), since it is already done internally by the ANN.

**Page 16, Line 2: Can you please be more explicit about what this matrix is, and to what matrix equation it is input into?**

Machine learning libraries work with matrices formed by a series of lines, one for each ground truth value used as a reference, and columns formed by the respective predictors for each ground truth value. Therefore, data generated following Eq. 4 is

structured as a matrix with the respective glacier-wide SMB values, forming a 1048x55 matrix. For each of the 1048 glacier-wide SMB values, we generate 55 predictors following Eq. 4.

In order to make this point clear, the whole paragraph has been rephrased as follows:

"32 glaciers over variable periods between 31 and 57 years result in 1048 glacier-wide SMB ground truth values. For each glacier-wide SMB value, 55 predictors were produced following Eq. 4: 33 combined predictors, with ${\Delta \overline{T}_{\operatorname{mon}}}$ and ${\Delta \overline{S}_{\operatorname{mon}}}$ accounting for 12 predictors each, one for each month of the year. All these values combined produce a 1048x55 matrix, given as input data to the OLS and Lasso machine learning libraries."

LaTex print: see previous image.
* * *
Page 16, Line 8: "...the annual CPDD as well as the winter and summer snowfall appear as significant predictors as well as several monthly mean temperatures and snowfall values..."; Change to "...the annual CPDD, winter and summer snowfall, and several monthly mean temperature and snowfall were found to be significant at p<?..."

In this sentence we did not imply to use the word "significant" in statistical terms. A more accurate word choice, in the context of the causal analysis which determines the importance (%) of each predictor would be:

"Regarding the climatic variables, the annual CPDD as well as the winter and summer snowfall appear as **the most important** predictors together with several monthly mean temperatures and snowfall values (Fig. 5)"
* * *
Page 20, Figure 8: This is a challenging plot to interpret (in my opinion). Can you plot deep learning bias vs lasso bias and deep learning MAE vs Lasso MAE? That may more clearly demonstrate the points you make, and may reveal structure. The points in the scatter plots could be coloured by region (Ecrins, Vanoise, Mont Blanc) if there are regional patterns. If this approach is not useful or helpful, then the current figure will suffice.

We agree that there is a lot of information in the plot, but after testing different types of plots, we believe this is the most effective way to convey the information. As requested, we plotted both the MAE and bias between Deep learning and Lasso, but in our opinion, these plots are not clear and do not serve to establish a clear comparison between the two methods.

[Figure]

[Figure]

The confusion might come from the fact that since glaciers are structured in massifs, the reader is tempted to look for regional patterns. Since there are no clear patterns, and the plot does not intend to show that, this now has specifically been mentioned in the legend as follows:

"Figure 8. Mean average error (MAE) and bias (vertical bars) for each glacier of the training dataset structured by clusters for the 1984-2014 LOGO glacier-wide SMB simulation. **No clear regional error patterns arise**"

Page 20, Line 15: "This implies, that . . ."; remove the comma

The sentence has been adapted as suggested by the reviewer.

Authors reply to Anonymous Referee #1's review on "Deep learning applied to glacier evolution modelling"
* * *
Page 22, Line 7: "...using Leave-Some-Glaciers-and-Years-Out (LSYGO)"; the abbreviation should be LSGYO, or the full phrase should be Leave-Some-Years-and-Glaciers-Out, for consistency
* * *
Indeed, the definition has been changed to "Leave-Some-Years-and-Glaciers-Out (LSYGO)".
* * *
Page 25, Lines 4-5: "The greater the dropout, the more we will constrain the learning of the ANN so the higher the generalization will be, until a certain point." This sentence is not clear. What does it mean to "constrain the learning"? Why is there a "certain point", and what happens beyond that point? This could be made more explicit.

As explained in "2.2.4 Deep artificial neural network", dropout is a regularization technique which consists of disconnecting certain connections in the neural network in order to reduce or constrain the amount of learning. This has been shown to help the ANN generalize. There is a range of dropout values which will produce this effect, but when pushed too far, the ANN will become too small, therefore unable to find meaningful patterns in data and performance will start to drop. The key aspect in the use of dropout is finding the good range of dropout values to make the ANN generalize without dropping performance. As with any hyperparameter tuning in ANNs, this is done via cross-validation as explained in the results section.

In order to make this point clearer, the sentence has been adjusted as follows:

"The greater the dropout, the more we will constrain the learning of the ANN so the higher the generalization will be, until a certain point, **where relevant information will start to be lost and performance will drop.**"
* * *
Page 25, Line 7: Why is it that slower convergence leads to better generalization? Is this always true?

For a given gradient descent optimizer, a slower convergence, meaning a smaller learning rate and a greater number of epochs, generally results in a better generalization. As explained in page 25, lines 6-7, a slower convergence has higher chances to encounter the global minima, whereas faster learning rates results in bigger jumps throughout the error landscape, thus often getting stuck in local minima or in certain regions of the error landscape. As for any hyperparameter, there is a range of values for which this is applicable. Depending on each case and dataset, a slower convergence might not improve generalization nor performance. Moreover, using a too slow converge will likely hamper or totally prevent any learning.

The sentence has been adjusted:

"On the other hand, the learning rate to compute the stochastic gradient descent, which tries to minimize the loss function, also plays an important role: smaller learning rates **generally** result in a slower convergence towards the absolute minima, thus producing models with better generalization."

Page 25, Line 8: "...that best suits a certain dataset and model." How does one define "best"?

The best model is determined using cross-validation and looking at different metrics, such as the RMSE or the coefficient of determination. The sentence has been updated to take this into account:

"By balancing all these different effects, one can achieve the accuracy versus generalization ratio that best suits a certain dataset and model **in terms of performance**"

Page 25, Line 17: "Despite it has been shown..."; Change to "Although it has been shown" or "Despite the fact that it has been shown"

The sentence has been modified to "Despite the fact that it has been shown…"

Page 25, Line 28: "The results were quite astonishing..."; If the results are astonishing, then this result warrants further emphasis in the paper. The methods used to come to this conclusion should be brought up in Section 3, and further discussion is warranted in Section 4. It is worth a figure to communicate these results

The adjective "astonishing" is probably not suitable in this context, as it might lead to exaggeration. These results are quite straightforward in terms of interpretation, since the methods are exactly the same as for the case study. The only difference is the number of input variables used. We believe that giving the numeric metrics is enough to convey the message, and such results do not deserve specific plots in an already quite long manuscript and supplementary material.

The adjective "astonishing" has been changed to "**interesting**" to reflect this.

Page 26, Lines 5-16: This paragraph is speculative, but is presented with a high degree of confidence. Phrases such as "unprecedented efficiency" and "excellent" are used without supporting evidence. Much of the discussion is implied; for example: "An interesting way of expanding a dataset would be to use a deep learning approach to fill the data gaps." It is unclear how this would be done. "Such an approach would be an excellent way of obtaining more SMB data in remote glacierized regions such as the Andes or the Himalayas." This is not known or demonstrated by the rest of the paper. I would recommend either removing this paragraph entirely or severely limiting its scope.

Indeed, this paragraph contains a lot of propositions, some not directly related to the results found in this paper. Some of the sentences have been removed to be more straight to the point and to avoid any speculation. Nonetheless, the first sentence "An interesting way of expanding a dataset would be to use a deep learning approach to fill the data gaps", is the direct consequence of all the methodology presented here. The relationships found and learnt in data can then be extrapolated to other glaciers and periods to make predictions. That is the main reason we have done such a thorough cross-validation in both the spatial and temporal dimensions. We have no studies to use as a reference for this, since our results of this regional SMB reconstruction will soon be submitted as a separate paper, and apparently there are

no other similar studies yet which have used such an approach. We believe this claims are valid, since the performance and viability of this approach is precisely proven in this methodological paper, for which we show the performance of this method compared to other classical approaches (multiple linear regression). Indeed, in other regions with smaller data coverage it might differ, but being in a discussion section, we think it is important to state the potential of this approach in these regions. Even if has been only tested in the European Alps for now, it would be extremely interesting to do so in regions such as the Andes or the High Mountains of Asia.

In order to deal with all these statements raised here, the paragraph has been widely updated:

"Deep learning can be of special interest once applied in the reconstruction of SMB time series. More and more SMB data is becoming available thanks to the advances in remote sensing sing (e.g., Brun et al. (2017); Rabatel et al. (2017); Zemp et al. (2019)), but these datasets often cover limited areas and the most recent time period in the studied regions. An interesting way of expanding a dataset would be to use a deep learning approach to fill the data gaps, **based on the relationships found in a subset of glaciers as in the case study presented here. Past SMB time series of vast glacierized regions could thereby be reconstructed, with potential applications in remote glacierized regions such as the Andes or the Himalayas**. ~~It could also be applied in data-rich regions benefiting from regionalized climate reanalyses (e.g. Caillouet et al. (2016)), covering the 1871-present period for France). Another possibility would be to completely bypass both the SMB and glacier dynamics of a classic glacier evolution model by training a deep ANN which would directly simulate changes in glacier thicknesses. If the ANN is trained with enough glacier thickness changes, climatic and topographical data, it could be able to simulate the 3D evolution of the glacier straight from the raw data. It might still be too soon for such models to be implemented, but once enough data will be available in the future, this could be a promising new way of tackling glacier evolution modelling.~~"
* * *
Page 26, Lines 18-22: This paragraph is speculative. It does not follow from the results presented in the paper, and is more of a justification for using deep learning in glaciology than it is an item of discussion in the context of the preceding research. These final two paragraphs do a disservice to the rest of the paper; prior to this, the organization had nice flow, and the first two paragraphs in Section 4.3 were both interesting and directly relevant.
* * *
We agree that this last paragraph is not directly related to the work presented here, and was included in the broader context of deep learning applied in glaciology. In order to keep the pace of the discussion and to simplify the discussion this whole last paragraph has been deleted from the manuscript.
* * *
Page 27, Lines 4-12: This paragraph is quite vague and does not explicitly follow from the research. For example: "It might still be too early for the development of such models in certain regions" is a vague statement. Conclusions should follow directly and explicitly from the work and should not reach beyond the scope of the research. The first paragraph in Section 5 is much better.
* * *
This paragraph focuses on the applications of the methodology presented here. As mentioned in a previous comment, these applications are just a consequence of the work and results presented in the French Alps case study. ALPGM, as a model and as a tool is capable of reconstructing and simulating glacier-wide SMBs at regional scale. We believe it is important to state why a model such ALPGM can be useful and what are its potential applications. These applications and their results will be presented as two separate papers, which show the results of applying deep learning for glacier-wide SMB reconstruction and for future glacier evolution predictions, using it as a SMB model (alternative to temperature index).

In order to improve the fluidity, and to relate all the statements to the research presented in this paper, the paragraph has been rephrased as follows:

"Deep learning should be seen as an opportunity by the glaciology community. Its good performance **for SMB modelling** in both the spatial and temporal dimensions shows how relevant it can be for a broad range of applications. Combined with in situ or remote sensing SMB estimations, it can serve to reconstruct SMB time series for regions or glaciers with already available data for past and future periods, **with potential applications in remote regions such as the Andes or the high mountains of Asia**. Moreover, deep learning can be used as an alternative to classical SMB models **as it is done in ALPGM**: important nonlinearities from the glacier and climate systems are potentially ignored by these mostly linear models, which could give an advantage to deep learning models in regional studies. It might still be too early for the development of such models in certain regions **which lack consistent datasets with a good spatial and temporal coverage**. **Nevertheless**, as new data becomes available the gap is slowly being closed towards real big data approaches in glaciology."

Supplementary Figures: In the SMB_lasso_ANN_no_weights_SMB_simulations.pdf file, y-axes are missing units.
The supplementary figures' y-axes have been updated as suggested by the reviewer.

Figures 6, 7, 9, and 11: Please increase font size, especially of axis labels.
The font size has been increased as suggested by the reviewer.

[revised manuscript text omitted]

---

## Author Response (AR2)

**Fabien Maussion**

This is my second assessment of Bolibar et al, 2019. I had several main concerns about the first version of the manuscript. Some of them were addressed by the authors, and the revised manuscript is much easier to follow and understand. I now also better understand the methods used in this study.

Some of my other concerns have been argued against, and resulted in no changes in the manuscript: mostly, my concern about targeting the prediction of glacier wide SMBs with the model. The argument used by the authors is based on "leave one year out" validation results and other predictor importance considerations which, I believe, do not allow to estimate the true effect of changing geometry on long (i.e. > decades) time scales. Furthermore, it must be noted that the "full SMB model" (predicting SMB based on *modelled* geometries) is only validated in Fig. 11, and only indirectly (via the area) i.e. without comparison with the reference SMB data (Figs. 6 to 10 still use known changing geometries, giving an advantage to the SMB model that it won't have outside the validation period).

Since this model is going to be used "as is" to provide future projections (and without starting an unnecessary flame war, this is still a nice paper), I would recommend the authors to mention these two points in the discussion section (maybe section in 4.3).

Finally, I spotted a few typos/errors which, I believe, will be corrected by Copernicus' language editing service.

Looking forward to see this paper published soon! Best,

Fabien Maussion

We are glad that the changes proposed by the reviewer allowed to improve the overall clarity of the manuscript.

Regarding the simulation of glacier-wide SMB, we understand the concerns of the reviewer. As it has been discussed in our previous reply, we agree that the simulation of glacier-wide SMB is more complex than point SMBs, which are independent from topography and glacier geometry. Nonetheless, we believe we have also made a strong point on why we chose to do so. In a data-driven approach data availability is crucial, so the fact that much more glacier-wide SMB data is available in the French Alps was a decisive factor. On top of that, we agree that this poses an extra difficulty in the validation of the full glacier evolution model, but this is not the only nor main application of this model. ALPGM is also being used to reconstruct past glacier-wide SMB series from observations for another paper, which is an application that has been thoroughly validated in this study (Fig. 6-10). Concerning the difficulties in validating the full glacier evolution with simulated topographies, we share the need with the reviewer to be clear about this in the paper, but it does not mean that such an approach is not feasible. We validated the model using simulated glacier geometries according to the methodological limitations established by data availability. Since glacier ice thickness estimations (Farinotti et al., 2019) date from the year 2003, our validation can only encompass the years 2003-2015, for which we have observations. As explained in our response to F. Maussion's first review, we validated for this period

both the simulation of glacier topographical parameters and the resulting glacier-wide SMB values. This was detailed in Sect. 3 in the Supplementary Material and in our reply to 1.1 from the reviewer's first report. The differences between simulated and observed topographical variables were very small, as shown in Table S1. Additionally, the predicted glacier-wide SMB using these simulated topographical parameters were compared to simulated glacier-wide SMB data using observations, as depicted in Fig. S6, with an average annual difference of 0.069 m.w.e. a$^{-1}$. We believe these results to be acceptable, but we also agree that the validation period is quite short to be representative of longer simulated periods (> decades).

Therefore, as suggested by the reviewer in this second assessment, we have edited some aspects in the discussion in order to be clear and transparent about this, as well as to take into account the reviewer's comments on the advantages of using point SMB *vs* glacier-wide SMB. Moreover, we have added some perspectives on the transition towards physics-informed data science approaches, in order to improve certain limitations of our current method. All changes are shown in bold in the updated 4.3 section shown below as well as some related changes in Sect. 5 Conclusions.

[revised manuscript text omitted]